# Research on the Relationship between the Structure of Forest and Grass Ecological Spaces and Ecological Service Capacity: A Case Study of the Wuding River Basin

**Yufan Zeng, Qiang Yu * , Xiaoci Wang, Jun Ma , Chenglong Xu, Shi Qiu , Wei Liu and Fei Wang**

College of Forestry, Beijing Forestry University, Beijing 100083, China; zengyufan59@bjfu.edu.cn (Y.Z.); wangxiaoci2118@bjfu.edu.cn (X.W.); majun_oitkg@bjfu.edu.cn (J.M.); xuchenglong@bjfu.edu.cn (C.X.); charles23@bjfu.edu.cn (S.Q.); vivian_liu@bjfu.edu.cn (W.L.); wangfei@bjfu.edu.cn (F.W.)
* Correspondence: yuqiang@bjfu.edu.cn; Tel.: +86-130-2123-9234

**Abstract:** In recent years, the accelerated pace of urbanization has increased patch fragmentation, which has had a certain impact on the structure and ecological environment of forest–grass ecological networks, and certain protection measures have been taken in various regions. Therefore, studying the spatiotemporal changes and correlations of ecological service functions and forest–grass ecological networks can help to better grasp the changes in landscape ecological structure and function. This paper takes the Wuding River Basin as the research area and uses the windbreak and sand fixation service capacity index, soil conservation capacity, and net primary productivity (NPP) to evaluate the ecological service capacity of the research area from the three dimensions of windbreak and sand fixation, soil conservation, and carbon sequestration. The Regional Sustainability and Environment Index (RSEI) is used to extract ecological source areas, and GIS spatial analysis and the minimum cumulative resistance (MCR) model are used to extract potential ecological corridors. Referring to complex network theory, topology metrics such as degree distribution and clustering coefficient are calculated, and their correlation with ecological service capacity is explored. The results show that the overall ecological service capacity of sand fixation, soil fixation, and carbon sequestration in the research area in 2020 has increased compared to 2000, and the ecological flow at the northern and northwest boundaries of the river basin has been enhanced, but there are still shortcomings such as fragmented ecological nodes, a low degree of clustering, and poor connectivity. In terms of the correlation between topology indicators and ecological service functions, the windbreak and sand fixation service capacity index have the strongest correlation with clustering and the largest grasp, while the correlation between soil conservation capacity and eigencentrality is the strongest and has the largest grasp. The correlation between NPP and other indicators is not obvious, and its correlation with eccentricity and eigencentrality is relatively large.

**Keywords:** forest–grass ecological network; complex network; MCR model; correlation analysis; ecological services

## 1. Introduction

In recent years, with the rapid development of the economy, the urbanization rate in China has increased year by year [1]. The increasingly fast urbanization process has changed the structure and function of land, causing a significant impact on the ecological environment [2,3]. The denser and more extensive transportation network has resulted in more fragmented biological habitats, reducing landscape connectivity [4]. In recent years, some scholars have proposed the concept and basic principles of an ecological spatial network from the perspective of landscape connectivity and the integrity of ecosystem structure and function [5]. The network is constructed by taking important ecological patches within the region as nodes and using strip corridors between patches as edges to mitigate the fragmentation of wildlife habitats and the decline in ecosystem services [6,7].

The forest and grass ecological spatial network is a type of ecological spatial network that increases the importance of researching forest and grass resources within the region. It is an important research topic in landscape ecology and a topic of great interest in other disciplines such as urban planning [8]. In recent years, related research has gradually developed towards multiple objectives, multiple functions, and multiple scales, covering more and more ecological services, and has become increasingly important in related fields [9,10]. The concept of forest and grassland ecological networks has not yet resulted in a consensus, and scholars have defined the networks from different research perspectives and goals, resulting in different network structures and various interpretations, but all emphasizing the consistency of ecological processes within the network [11]. Additionally, connectivity, functionality, and integrity are also essential characteristics [10].

The complete ecological spatial network includes three important components: ecological source areas, ecological corridors, and ecological nodes. The extraction methods for ecological source areas mainly include direct identification, index evaluation, morphological spatial pattern analysis (MSPA), and comprehensive identification methods. In this paper, the comprehensive identification method is adopted, which integrates the advantages of the other three methods, such as the intuitive and easy-to-understand nature of the direct recognition method [12], the objective quantification of the index evaluation method [13], and the precise identification of landscape characteristics provided by MSPA [14]. This method can better reflect the role of ecological source areas in both landscape structure and ecological function. This method is now being accepted and used by more and more scholars [10,15,16]. Ecological corridors, as the backbone of the ecological spatial network, are also essential to extract [17]. Ecological corridors are usually extracted using the MCR model or circuit theory. The MCR model mainly considers three elements—source areas, resistance surfaces, and cumulative costs—and constructs ecological corridors by extracting the minimum resistance path of ecological flows between source areas. Due to its simple structure and clear algorithm, it has become the mainstream method for extracting ecological corridors [18–20]. KNAAPEN [21] first proposed the MCR model in 1992 for constructing ecological corridors and used it to simulate animal migration, but it was still subject to many constraints. Yu [22] first applied the model to landscape pattern analysis in 2012, providing a new approach for future studies of landscape patterns. Su [23] constructed the forest and grass ecological network (FG ecological network) in Dengkou County, Inner Mongolia in 2019 and analyzed the topological structure and statistical characteristics of the FG ecological network under 11 development scenarios using complex network theory. They quantified the attribute information of the ecological spatial network, providing a way to quantitatively describe ecological spatial networks. Huang [24] used landscape pattern indices to analyze landscape features, identified ecological sources, landscape connectivity, and regional attribute-corrected ecological resistance surfaces using MSPA, and constructed an urban ecological network using the MCR model to improve landscape connectivity. The limitation of this study is that MSPA cannot fully consider the role of ecological function when identifying ecological source areas. Yang [25] constructed a forest and grass ecological spatial network in China in 2022 based on the MCR model, calculated the topological indicators of the ecological spatial network using complex network theory, and explored the relationship between the topological structure of the forest and grass ecological spatial network and ecosystem services in China. The study is more comprehensive and advanced, and all aspects are considered, but the index evaluation method is used in the extraction of source areas, and the ecological function and landscape structure can be better combined for extraction.

Scholars have found that by introducing complex network theory into ecospatial networks, the functions and indicators of ecological spatial networks can be better quantitatively described by calculating the topological indicators of the network [4,21,23]. The study of complex network theory can be traced back to the "Seven Bridges of Königsberg" problem proposed by the mathematician Euler in the 18th century. Euler abstracted the blocky land as points and the bridges connecting the land as edges, and the points and

edges connecting points constituted a network; however, there is still no unified and clear definition of complex networks. SH Strogatz [26] proposed in 2001 that the most fundamental problem of complex networks is the structural problem and conducted research on the topological structure and dynamics of complex networks, which provided a good theoretical basis for subsequent research based on complex networks. G Csardi [27] introduced a software package in 2006, which greatly simplified many of the more complex calculations and facilitated research on complex networks. SN Dorogovtsev [28] reviewed the achievements, concepts, and methods in the development of new critical phenomena in complex networks in 2008, believing that a critical phenomenon is a key phenomenon in complex networks, and discussed critical phenomena in different categories of systems and further improved the theoretical framework of complex networks. Gao [29] used a network evaluation system to study the changes of complex networks over time, which provides a good theoretical guidance for studying the changes of network structure in time series. Yang [4] combined complex network theory and ecospatial networks to quantitatively optimize the ecospatial network in the Songhua River Basin from a theoretical perspective, so that the research results were more objective and intuitive, and good results were achieved.

The Wuding River, as a primary tributary of the Yellow River and also the largest river in the Yulin region of Shaanxi Province, has ecological problems that affect the green and sustainable development of the Yulin region. Moreover, the Wuding River Basin is located in the transitional zone between semi-arid and arid areas and within the scope of the Loess Plateau, where soil erosion is severe, which makes it an area with typical geographic features. There have been few studies on the temporal and spatial changes of the Wuding River Basin, and few scholars have used complex network theory to explore the changes in the forest and grassland ecological space of the Wuding River Basin. Furthermore, there is no research on the combination of the topological indices of the ecological spatial network in this area with its ecological service capacity. By studying the forest and grassland ecological network and its relationship with ecological service functions, the trend of ecological change in the Wuding River Basin can be determined, the reasons for the change can be analyzed, and more appropriate recommendations can be proposed to respond to environmental changes.

This article focuses on the Wuding River Basin and uses data from four periods (2000, 2010, 2015, and 2020) to construct a forest and grassland ecological network and evaluate changes in the ecological services of sand fixation, soil conservation, and carbon sequestration. Based on the theory of complex networks, the study calculates the topological indicators for each year and explores their changes in spatiotemporal sequence. Using the results from 2020 as an example, the article investigates the correlation between topological indicators and ecological services. The study area is geographically important and representative, but there has been little research conducted in this region. To address this gap, the article innovatively uses a comprehensive identification method to extract and select ecological source areas, which takes into account both landscape structure and ecological function. The article also fills the gap in previous research on ecological spatial networks that relied on direct extraction methods or index evaluation methods. The article employs GIS spatial analysis and MCR models to extract potential ecological corridors, which have a simple structure and clear algorithm. Using the theory and methods of complex networks, the study calculates topological indicators such as degree and degree distribution, average path length, and clustering coefficient, which quantitatively and intuitively reflect the structure and functional characteristics of the forest and grassland ecological spatial network. Exploring the relationship between ecological service functions and topological indicators of ecological networks can provide a scientific basis and decision support for ecological restoration in the study area. Overall, this article partially addresses the shortcomings of previous research and has some innovation.

## 2. Materials and Research Methods

This article employs the RSEI index to preliminarily extract ecologically significant areas and utilizes the digital elevation model (DEM), slope, normalized vegetation index (NDVI), normalized difference water index (NDWI), and soil cover type data to derive cumulative resistance surfaces for constructing a forest and grassland ecological spatial network. The various topological indices of the forest and grassland ecological spatial network are calculated, and their correlations with ecological service capacity indices (sand fixation, soil conservation, and carbon sequestration) are analyzed. The data processing flow and idea of this paper are shown in Figure 1.

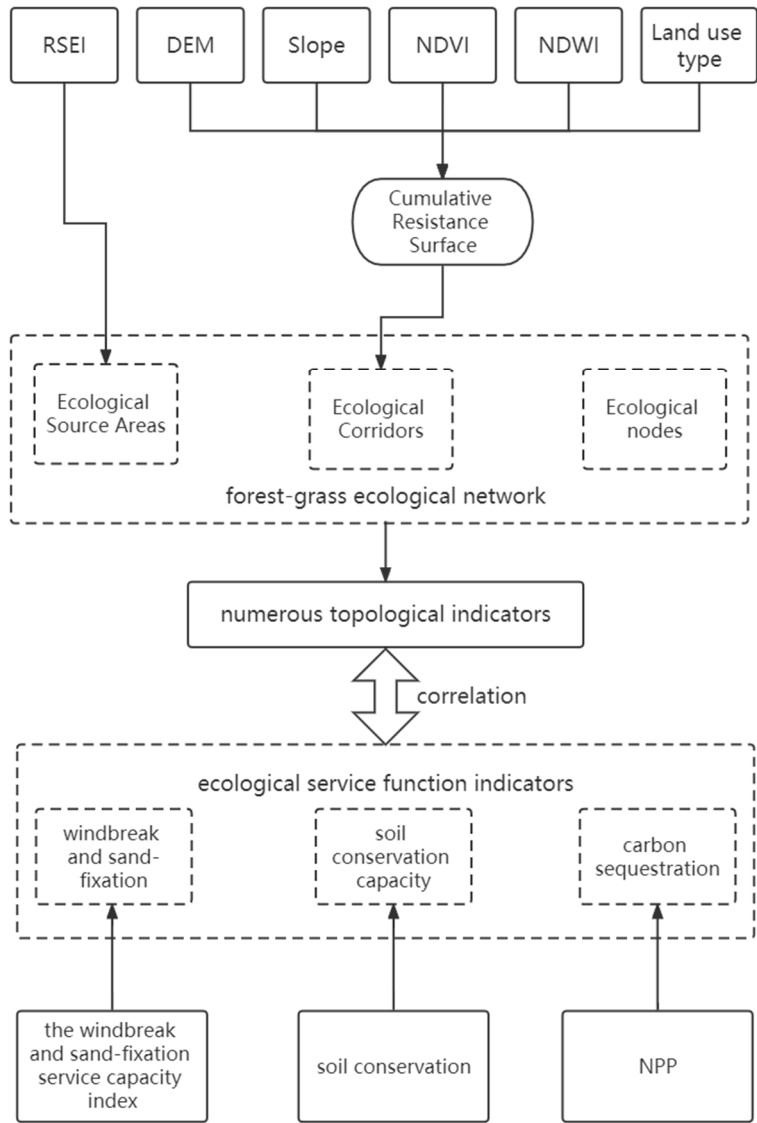

**Figure 1.** Data processing flow.

### 2.1. Overview of the Study Area

The study area is located in the hinterland of the "Ji"-shaped Yellow River, which is an important tributary of the middle Yellow River, with 70.3% of the study area located in the northern part of Shaanxi Province and 29.7% located in the southwest of Inner Mongolia Province. Its geographical location is between 37°–39°N and 108°–111°E, with a total basin area of approximately 30,300 km$^2$ (of which the area within Shanxi Province is approximately 21,000 km$^2$, and the area within Inner Mongolia Province is approximately 9300 km$^2$). It is located in the transitional zone between the semi-arid and arid regions, and

the climate is classified as temperate and subtropical, belonging to the temperate continental climate zone. The annual average temperature ranges from 6 to 12 °C, and the annual precipitation ranges from 270 to 660 mm, with an annual evaporation of 1268 to 1457 mm. The main land use type in the study area is grassland, and the sum of forest and grass land use types accounts for about 78% of the total area. Among them, forest land with canopy density greater than 30 accounts for more than 50% of the total forest area, and grassland with medium-high coverage (over 20% coverage, better water conditions, and dense grass cover) accounts for about 39% of the grassland. The location map of the study area is shown in Figure 2.

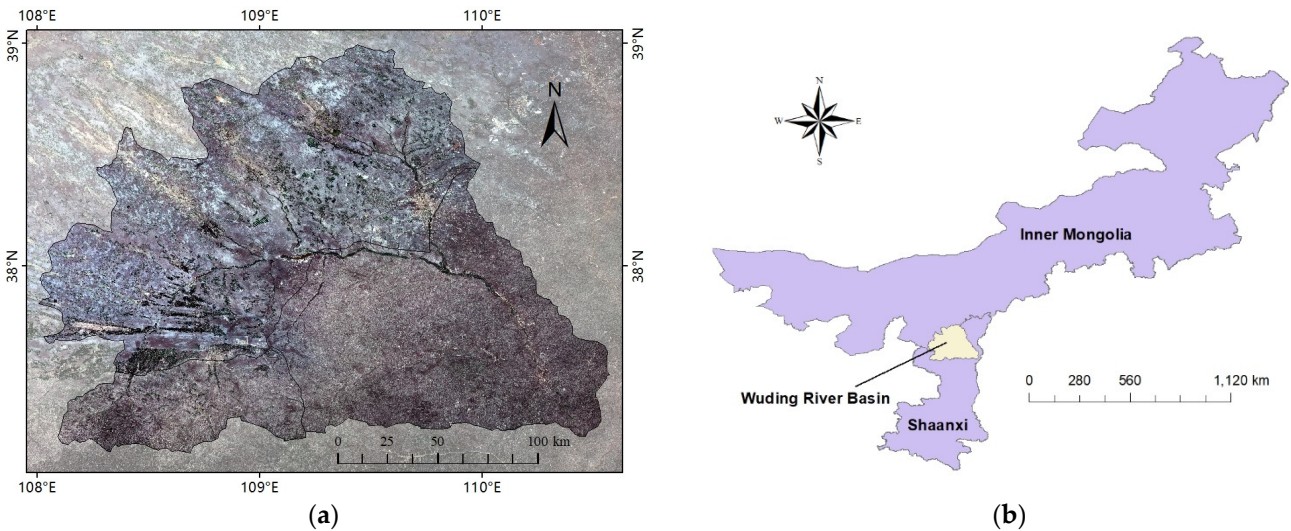

(**a**)  (**b**)

**Figure 2.** Overview of the study area (**a**) and geographical location (**b**).

### 2.2. Data Sources and Processing

The DEM data for the study area were obtained from the GDEMV3 digital elevation data product of the Geographic Spatial Data Cloud (http://www.gscloud.cn/, accessed on 14 December 2022), with a spatial resolution of 30 m. The slope data were derived from DEM using slope analysis in ArcGIS. The land-use data were obtained from the China Land Cover Data Set (CLCD) [30] created by Yang Jie and Huang Xin, with a spatial resolution of 30m. The normalized difference NDVI and NDWI data were acquired from the Landsat5 (2000, 2010) and Landsat8 (2015, 2020) data sets on the Google Earth Engine platform (http://code.earthengine.google.com/, accessed on 8 December 2022), with a spatial resolution of 30 m. The vegetation NPP data were obtained from the MODIS MOD17A3 data set. The heat index data were acquired from the Landsat LST data set, with a spatial resolution of 30 m. The annual mean temperature, annual precipitation, and annual evaporation data were obtained from the Resource and Environment Science and Data Center of the Chinese Academy of Sciences (http://www.resdc.cn/, accessed on 20 December 2022). Data processing was conducted using platforms such as ENVI 5.3, Google Earth Engine, ArcGIS 10.0, Matlab, Gephi, and Fragstat4.2.

### 2.3. Calculation of Ecological Remote Sensing Indices

The RSEI, proposed by Xu Hanqiu [31], is a remote sensing-based index mainly composed of natural factors used for the rapid monitoring and evaluation of the ecological environment. The RSEI integrated the greenness, heat, wetness, and dryness indices using principal component analysis, and each index was calculated using Landsat bands. The greenness index was calculated using the red and near-infrared spectra as follows:

$$NDVI = (NIR - Red)/(NIR + Red)$$

The heat index generally refers to land surface temperature (*LST*), which was represented using temperature after being corrected for emissivity and atmospheric transmittance:

$$Ts = T_b / [1 + (\lambda T / \alpha) ln\varepsilon]$$

$$LST = K_2 / ln \left[ K_1 B(Ts) + 1 \right]$$

where $K_1$ and $K_2$ were 607.76 and 1260.56, respectively, for Landsat 5 images, and 774.89 and 1201.14, respectively, for Landsat 8 images [32]. The wetness index (WET) was obtained by summing the product of each band's sensor parameter and band value. The parameters $c_1$–$c_6$ were 0.0315, 0.2021, 0.3012, 0.1594, −0.6806, and −0.6109, respectively, for the TM sensor of Landsat 5, and 0.1511, 0.1973, 0.3283, 0.3407, −0.7117, and −0.4559, respectively, for the OLI sensor of Landsat 8. The dryness index was synthesized using the bare soil index (*SI*) and the built-up index (*IBI*) and expressed as *NDBSI*, with Landsat 8 images as an example:

$$NDBSI = \frac{SI + IBI}{2}$$

$$IBI = \frac{\left\{ \frac{2B_5}{B_5 + B_4} - \left[ \frac{B_4}{B_3 + B_4} + \frac{B_2}{B_2 + B_5} \right] \right\}}{\left\{ \frac{2B_5}{B_5 + B_4} + \left[ \frac{B_4}{B_3 + B_4} + \frac{B_2}{B_2 + B_5} \right] \right\}}$$

$$SI = [(B5 + B3) - (B4 + B1)] / [(B5 + B3) + (B4 + B1)]$$

After normalizing the above four indicators and using ENVI to create the first principal component (PC1) as the RSEI, the calculated results were standardized to obtain the final outcome. The RSEI ranges from 0 to 1, with a higher value indicating a better ecological condition in the area and a lower value indicating poorer ecological environmental quality.

$$RSEI_0 = 1 - 0.0708 NDVI - 0.0076 WET - 0.0053 LST - 0.0002 NDBSI$$

$$RSEI = (RSEI_0 - RSEI_{0min}) / (RSEI_{0max} - RSEI_{0min})$$

### 2.4. Ecological Source Site Extraction

Ecological source sites refer to areas that can provide material and energy to adjacent ecosystems, as well as organic sources for organisms, and can be used for material transfer by various plants and animals, promoting the energy flow of the ecosystem. The RSEI index is based on remote sensing image calculations and primarily represents the natural ecological environment quality of the region, which can effectively indicate the ecological source site. Based on a literature review, we obtained RSEI values from arid or semi-arid regions with similar characteristics to the study area. The research showed that the RSEI value for the Shaanxi plateau was around 0.44, and the RSEI value for the Gansu plateau was around 0.45 [33]. In the humid area, the RSEI value for the Erhai Lake basin was around 0.51 in recent years [34]. This study used these values as reference to assess the RSEI value for the Wuding River basin and determined that areas with RSEI values greater than 0.46 are considered ecologically better regions. In this study, areas with high RSEI values (RSEI > 0.46) were selected as source sites, and then the ecological source site was filtered based on three indicators: patch area, average NDVI, and average NDWI, which were calculated using the ArcGIS raster calculator, resulting in the final ecological source site.

### 2.5. Construction of Forest and Grass Ecological Network Resistance Surface

This study selected five main ecological factors, including elevation, slope, NDVI, NDWI, and land use data. According to their definitions, they belong to four categories: topography, vegetation cover, hydrological distribution, and land cover, as the reference indicators of these four aspects. Figure 3 shows the five ecological factors mentioned above for 2020; pictures of the ecological factors for the remaining years are attached in the

annex. The natural breakpoint method was used to reclassify these five ecological factors in ArcGIS. Referring to natural laws and previous research results [4,35–37], resistance coefficients were set for these five ecological factors and divided into five levels, with values of 1, 3, 5, 7, and 9. The ecological resistance surface of the research area was constructed by adding these values together. The larger the value, the greater the resistance value, and the more the energy flow is blocked; the smaller the value, the more favorable the energy flow [20,38]. The resistance coefficients for each ecological factor are shown in Table 1. The ecological resistance factors in 2010, 2015 and 2020 are shown in Figures S1–S3.

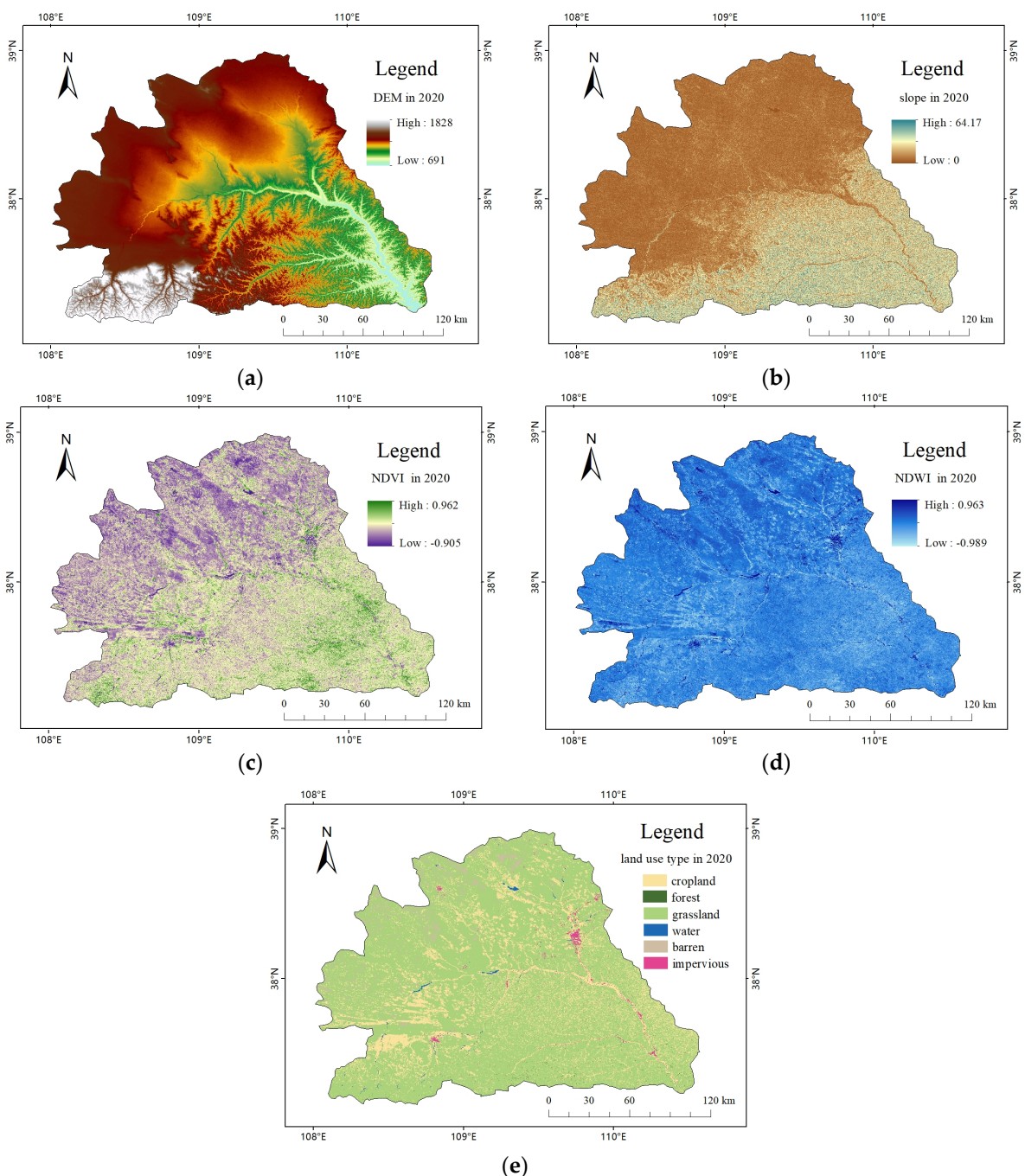

**Figure 3.** Ecological resistance factors of the study area in the year 2020. (**a**) DEM in 2020, (**b**) Slope in 2020, (**c**) NDVI in 2020, (**d**) NDWI in 2020, (**e**) Land use type in 2020.

**Table 1.** Classification and value of ecological resistance factors.

| Primary Impact Factor | Secondary Impact Factor | Classification | Resistance Value |
|---|---|---|---|
| Topography | DEM (m) | <900 | 1 |
| | | 900–1100 | 3 |
| | | 1100–1300 | 5 |
| | | 1300–1500 | 7 |
| | | >1500 | 9 |
| | Slope (°) | <3 | 1 |
| | | 3–9 | 3 |
| | | 9–18 | 5 |
| | | 18–27 | 7 |
| | | >27 | 9 |
| Vegetation cover | NDVI | <0 | 9 |
| | | 0–0.2 | 7 |
| | | 0.2–0.4 | 5 |
| | | 0.4–0.6 | 3 |
| | | 0.6–1 | 1 |
| Hydrological distribution | NDWI | <0 | 9 |
| | | 0–0.3 | 7 |
| | | 0.3–0.6 | 5 |
| | | 0.6–0.8 | 3 |
| | | 0.8–1 | 1 |
| Land cover | Land use type | Cropland | 5 |
| | | Forest | 1 |
| | | Grassland | 1 |
| | | Water | 1 |
| | | Barren | 7 |
| | | Impervious | 9 |

*2.6. Extraction of Potential Ecological Corridors*

Cumulative cost refers to the work done or resistance overcome when moving from one ecological source site to another adjacent ecological source site through different landscape units. Ecological corridors can be regarded as the collection of minimum cumulative costs between different ecological source sites, which can serve as channels to communicate and connect relatively isolated ecological units, typically presenting linear or belt-like distributions, and facilitating the flow of energy in the ecosystem. When energy flows through ecological corridors, the least amount of energy is lost, resulting in maximum ecological benefits [7]. The construction and extraction of ecological corridors generally use the MCR model, which is based on three elements: source sites, resistance surfaces, and accumulated costs. This model constructs the path of least resistance that enables ecological flows to move between adjacent source sites, thereby forming ecological corridors. Based on the MCR model, this study used the extracted source sites and constructed resistance surfaces to calculate the path of minimum energy consumption between adjacent source sites for each year in the Wuding River Basin, using Cost-path in ArcGIS and Matlab to obtain the potential ecological corridors.

*2.7. Ecological Service Function Indicators*

2.7.1. Windbreak and Sand-Fixation Capability

Windbreak and sand fixation are important ecological system functions in the arid and semi-arid regions of northern China. Scientific evaluation of windbreak and sand-fixation functions can provide a scientific basis and guidance for the protection and restoration of the ecological environment and the delineation of ecological protection redlines [39]. This

study used the index of windbreak and sand fixation service capacity of the ecosystem as the evaluation indicator, and its calculation formula is

$$Sws = NPP_{mean} \cdot K \cdot Fq \cdot D$$

Here, $NPP_{mean}$ refers to the mean value of net primary productivity of vegetation over many years, $K$ represents soil erodibility, $D$ stands for surface roughness, and $Fq$ is the average climatic erosivity factor over many years. A larger value of these factors indicates better windbreak and sand-fixation services, as well as a greater amount of sand fixation.

### 2.7.2. Soil Conservation Capacity

Soil erosion was initially proposed by foreign scholars from a geological perspective to describe the levelling effect of external forces [40,41]. In most Chinese literature, soil erosion is defined as the process of stripping, damaging, dispersing, separating, transporting, and depositing soil or other ground materials under the action of external forces. Soil erosion causes the loss of topsoil and deterioration of soil quality in the eroded areas [42–44]. Soil conservation is an important ecosystem service provided by the ecosystem to reduce soil erosion, which plays an important role in protecting the ecological environment of the eroded area and maintaining sustainable economic development [45]. This study used soil conservation quantity as the evaluation index of soil conservation capacity in the study area, and the revised universal soil loss equation (RUSLE) model was used to calculate it, with the following calculation formula:

$$A = R \cdot K \cdot LS \cdot C \cdot P$$

Among them, $A$ is the predicted amount of soil erosion; $R$ is the rainfall erosion factor, obtained from the precipitation data; $K$ is the soil erosion factor, which is obtained from the soil data; $LS$ is the slope length slope factor, calculated with a DEM; $C$ is the coverage and management factor; and $P$ is the soil and water conservation measure factor, which is obtained from remote sensing image data and land use data. Soil conservation is calculated in tons, and the higher the number, the higher the soil conservation capacity of the area.

### 2.7.3. Carbon Sequestration Capacity

The NPP of vegetation refers to the organic matter produced by producers in the ecosystem through photosynthesis by absorbing $CO_2$ from the atmosphere and undergoing physiological and biochemical processes in the ecosystem [46,47]. It is the remaining part of the total organic matter produced by producers through photosynthesis after deducting the organic matter consumed by respiration [48,49]. NPP is an extremely important manifestation of the carbon cycle process and is also the most direct and significant indicator of vegetation carbon sequestration capacity [50–52]. This study used NPP as the evaluation index of carbon sequestration capacity in the study area.

### *2.8. Basic Characteristics of Complex Networks*

### 2.8.1. Degree and Degree Distribution

In complex network structures, the degree of a node refers to the number of edges connected to the node. In an undirected graph $G = G(V, E)$, the degree of a vertex $v_{i_0}$ can be represented as

$$\mathrm{d}\left(v_{i_0}\right) = \sum_{i=i_0} e_{i,j}$$

The average value of the degrees of all nodes in a network is defined as the average degree, and the distribution of degrees for each node is defined as the degree distribution. Applying the concept of complex networks to the forest and grassland ecological network, the ecological nodes in the forest and grassland ecological network correspond to the nodes in the network, and the degree of an ecological node refers to the number of potential ecological corridors connected to it. Since the number of ecological corridors connected to

each ecological node varies, the corresponding degree sizes are also different. The greater the degree of an ecological node, the higher its importance in the forest and grassland ecological network. The average degree of a specified area reflects the importance of potential ecological nodes in that area.

### 2.8.2. Average Path Length

The distance between two nodes in a network is defined as the number of edges on the shortest path connecting them, and when two nodes are not connected, the distance is taken as positive infinity. The network diameter is defined as the maximum distance between any two nodes in the network [53,54]. The average path length of a network is the average of the shortest path lengths between any two nodes, reflecting the smoothness of energy flow in the forest and grassland ecological network. When the path length is too large, maintaining the forest and grassland ecological network will consume more energy and cause unnecessary energy loss [55].

### 2.8.3. Clustering Coefficient

The clustering coefficient reflects the degree of connection between a node and its neighboring nodes. The clustering coefficient ranges from 0 to 1, with a larger value indicating a stronger connection and more frequent communication between nodes [56,57]. When the value is 0, the edge does not exist. The larger the clustering coefficient of a network, the more evenly the nodes are dispersed and distributed in the network; that is, the larger the clustering coefficient of potential ecological nodes in the forest and grassland ecological network, the more evenly the ecological nodes are distributed.

## 3. Results and Analysis

### 3.1. Ecological Service Function Indicators

### 3.1.1. Windbreak and Sand-Fixation Capacity

Figure 4 shows the change in windbreak and sand fixation capacity in the study area from 2000 to 2020. In 2000, the region with the highest ecological system windbreak and sand fixation service index was in the northern part of the watershed, and the index in the central part was also relatively high, while the values in the east and west were lower, with the southeast corner having the lowest value. In 2010, the overall distribution of values was similar to that in 2000, and the average value of the entire region increased slightly. In the area with higher values in the north in 2000, there were small areas where the values decreased, but the lowest value in the watershed increased significantly. In 2015, the average value increased slightly compared to 2010, but the overall distribution change was not significant, and the maximum value decreased slightly while the minimum value increased slightly. In 2020, the minimum value increased significantly compared to 2015, the maximum value increased slightly, and the overall average value increased slightly. From 2000 to 2020, the distribution pattern of the windbreak and sand fixation service index of the ecosystem in the Wuding River Basin was similar, showing a trend of higher values in the north and lower values in the south, higher values in the middle, and lower in the east and west, with a slightly semi-enclosed shape. The average value of the index increased year by year, indicating that the wind and sand fixation service capacity of the ecosystem in the watershed has been increasing year by year. In terms of regional division, the values in the Heliangjian area of Heyuan were generally low, indicating poor windbreak and sand fixation service capacity, while the values in the loess hill and gully area were higher in the west and lower in the east. Windbreak and Sand Fixation in the Study Area from 2000 to 2020 are shown in Figure S4.

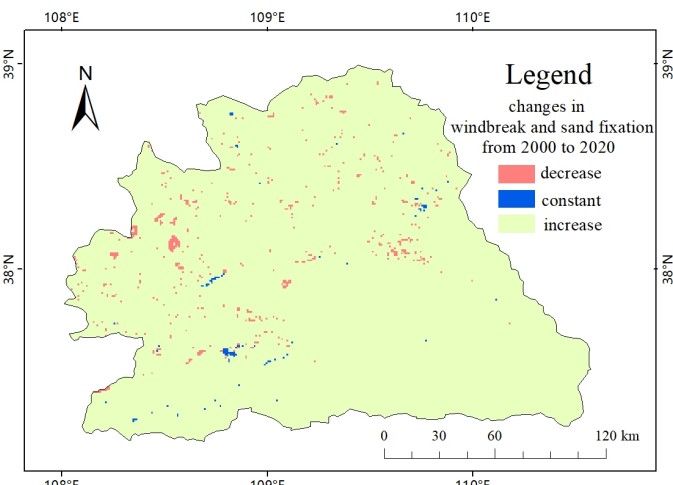

**Figure 4.** Changes in windbreak and sand fixation in the study area from 2000 to 2020.

### 3.1.2. Soil Conservation Capacity

Figure 5 shows the changes in soil conservation from 2000 to 2020. The mean soil conservation capacity within the basin was 5643.58 tons in 2000, 6390.58 tons in 2010, 6299.52 tons in 2015, and 6556.28 tons in 2020. Although there were fluctuations, soil conservation capacity generally showed a certain degree of improvement. From 2000 to 2020, the distribution of soil conservation capacity values was very similar in size. Regions with higher values were mostly distributed near the southern boundary, showing a general pattern of decreasing from south to north, and roughly distributed as high in the southeast and low in the northwest. In terms of regional division, the soil conservation capacity in the loess hill and gully area and the river-source ridge and valley area was strong, while the soil conservation capacity in the wind and sand area was very weak. Soil Conservation Amount of the Study Area from 2000 to 2020 are shown in Figure S5.

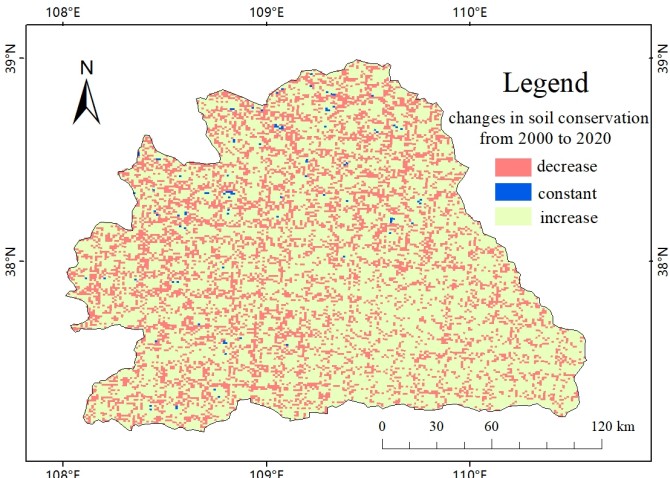

**Figure 5.** Changes in soil conservation amounts of the study area from 2000 to 2020.

### 3.1.3. Carbon Sequestration Capacity

Figure 6 shows the changes in NPP from 2000 to 2020. The distribution of NPP values within the basin showed a decreasing trend from southeast to northwest, with the Wuding River system and the southeast and southwest corners being the highest areas. The NPP value in the loess hill and gully area decreased from east to west, with generally high values and strong service capacity. The NPP value in the river-source ridge and valley area was low in the west and high in the middle and east, with a large overall value and good carbon sequestration effect. The NPP value in the wind and sand area, except for the areas around

the water system, was generally low, with a poor carbon sequestration effect. NPP of the Study Area from 2000 to 2020 are shown in Figure S6.

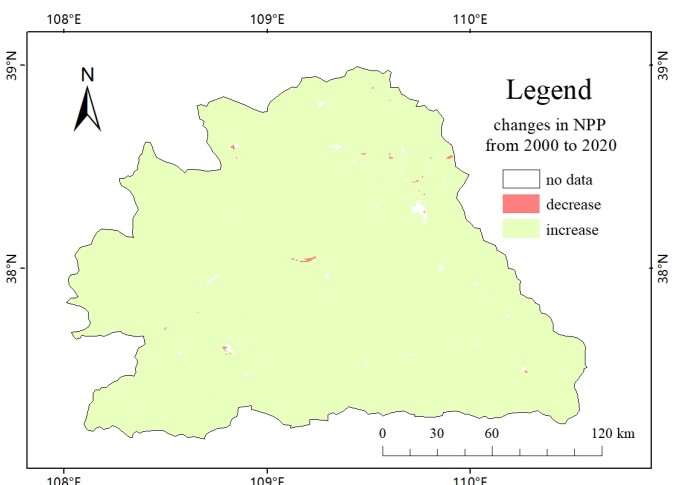

**Figure 6.** Changes in NPP of the study area from 2000 to 2020.

*3.2. Potential Ecological Network of Forests and Grasslands*

After extracting and screening the ecological source areas, distribution maps of ecological source areas were obtained for each year. In 2000, a total of 418 source areas were extracted, covering an area of 712 pixels. In 2010, 423 source areas were extracted, covering an area of 337 pixels. In 2015, 433 source areas were extracted, covering an area of 415 pixels. In 2020, 415 source areas were extracted, covering an area of 346 pixels. The total area of ecological source areas was the largest in 2000, mainly distributed along the Wuding River water system, mostly in the central and eastern parts of the basin, with a relatively continuous distribution. There were obvious bifurcations in the central part along the direction of the water flow, while other scattered areas were distributed in the southwest, north, and southeast parts of the basin. The larger source areas were mainly located in a belt-shaped region in the central and eastern parts, which belonged to the loess hill and gully region and were consistent with the real situation and natural laws.

In 2010, the source areas were still mainly distributed along the Wuding River water system, but the distribution was more scattered than that in 2000. In 2015, the source areas were mainly distributed in the central and eastern parts of the basin. Compared with 2010, the number and total area of source areas increased, and the overall distribution became more scattered, but there were some densely distributed areas in each small patch, without an obvious trend of distribution along the water system.

In 2020, source areas were distributed in all parts of the basin, mainly concentrated in the southeast and north, with three patches in the northeast–southwest direction in the southeast, belt-shaped distribution in the northwest–southeast direction in the north, and scattered distribution in other areas without an obvious trend of distribution along the water system. Compared with 2015, the number and total area of source areas decreased, and the distribution became more scattered. The number of source areas decreased in the central and southern parts and increased in the central-western and northern parts. Compared with 2000, the total number and area of source areas decreased in 2020, with a total area of only 48.6% of that in 2000, and the average area of each source area significantly decreased. The distribution pattern of source areas was no longer along the path of the Wuding River water system, and the overall distribution was more scattered and uniform. From 2000 to 2020, the distribution of source areas showed a pattern of more numerous and larger patches in the loess hill and gully region, and smaller and scattered patches in the sandy area, which was consistent with the natural conditions and cognitive experience. The distribution of ecological sources from 2000 to 2020 is shown in Figure 7.

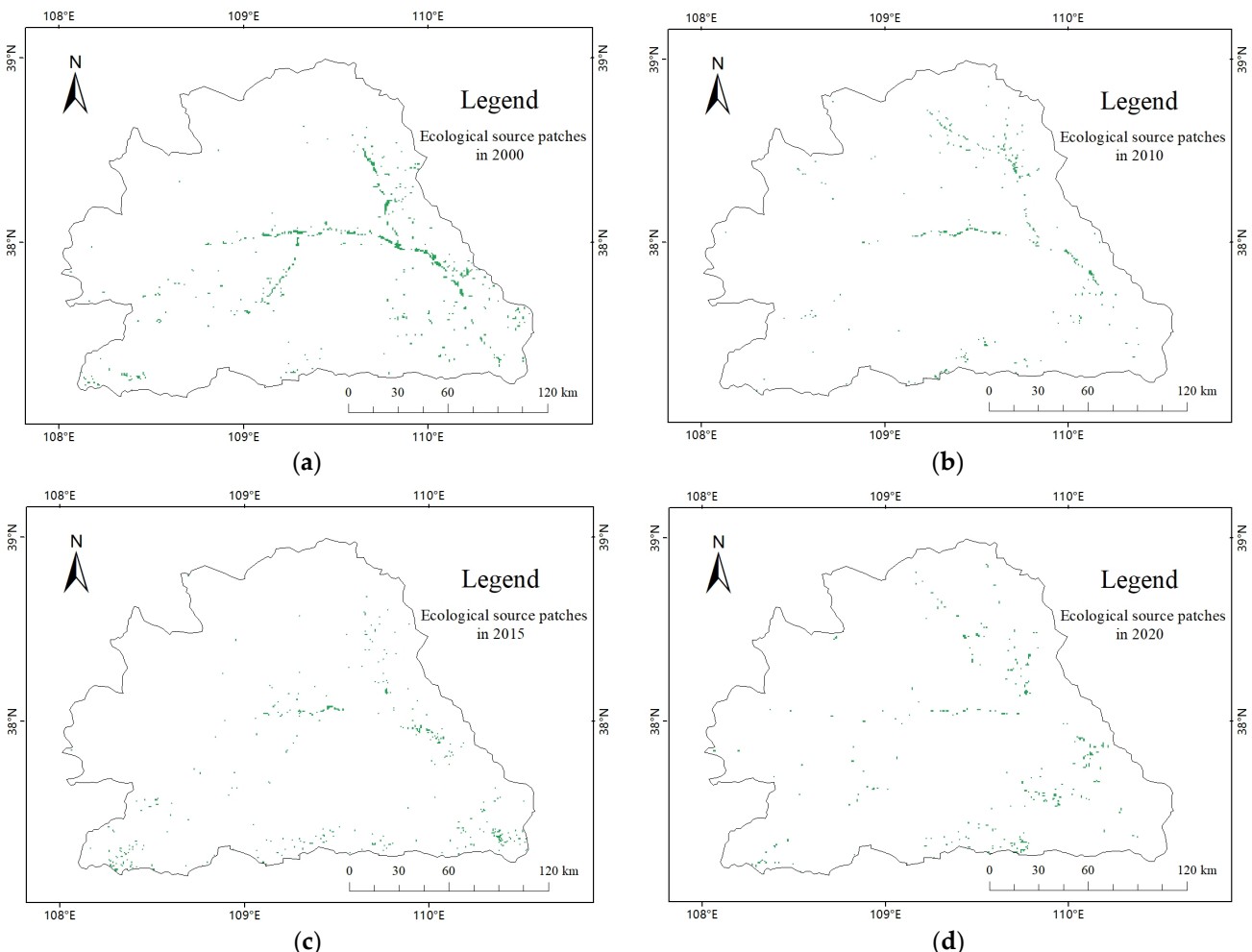

**Figure 7.** Distribution of ecological source areas in the study area from 2000 to 2020. (**a**) Ecological source patches in 2000, (**b**) Ecological source patches in 2010, (**c**) Ecological source patches in 2015, (**d**) Ecological source patches in 2020.

An overlay analysis was conducted on five ecological factors to construct annual ecological resistance surfaces. In 2000, the minimum resistance values were located along the Wuding River system and distributed in the central and eastern regions as the river flowed. The resistance value in the main channel was lower than that in the tributaries, and the resistance value in the network-like tributary basin was relatively small. The overall resistance value in the central and southern regions was high, while the southwest corner had the highest resistance value due to its high elevation and distance from the water system, which had less influence from water source factors. In 2010, the distribution of resistance values was similar to that in 2000, but the overall resistance value increased, mainly in the central and southern regions, the network-like tributary basin, and the southwest corner. In 2015, compared to 2010, the most significant change in the distribution of resistance values was the overall decrease in resistance value in the network-like tributary basin and the overall increase in resistance value in the adjacent areas of the southeast water system, while the range of the region with high resistance values in the central and southern regions slightly expanded. In 2020, compared to 2015, the overall resistance value slightly decreased, mainly reflected in the overall slight decrease in resistance value in the tributary basin and the slight reduction in the range of areas with high resistance values in the central and southern regions.

From 2000 to 2020, the overall distribution pattern of resistance values within the basin remained similar, with no significant changes. The areas with low resistance values were

concentrated along the main channel and around the network-like tributary basin, while the high resistance value area was in the source area of the river valley, and the resistance value in the sandy area was relatively low. The overall resistance value in the loess hill and gully area was high in the west and low in the east. The distribution of cumulative resistance values from 2000 to 2020 is shown in Figure 8.

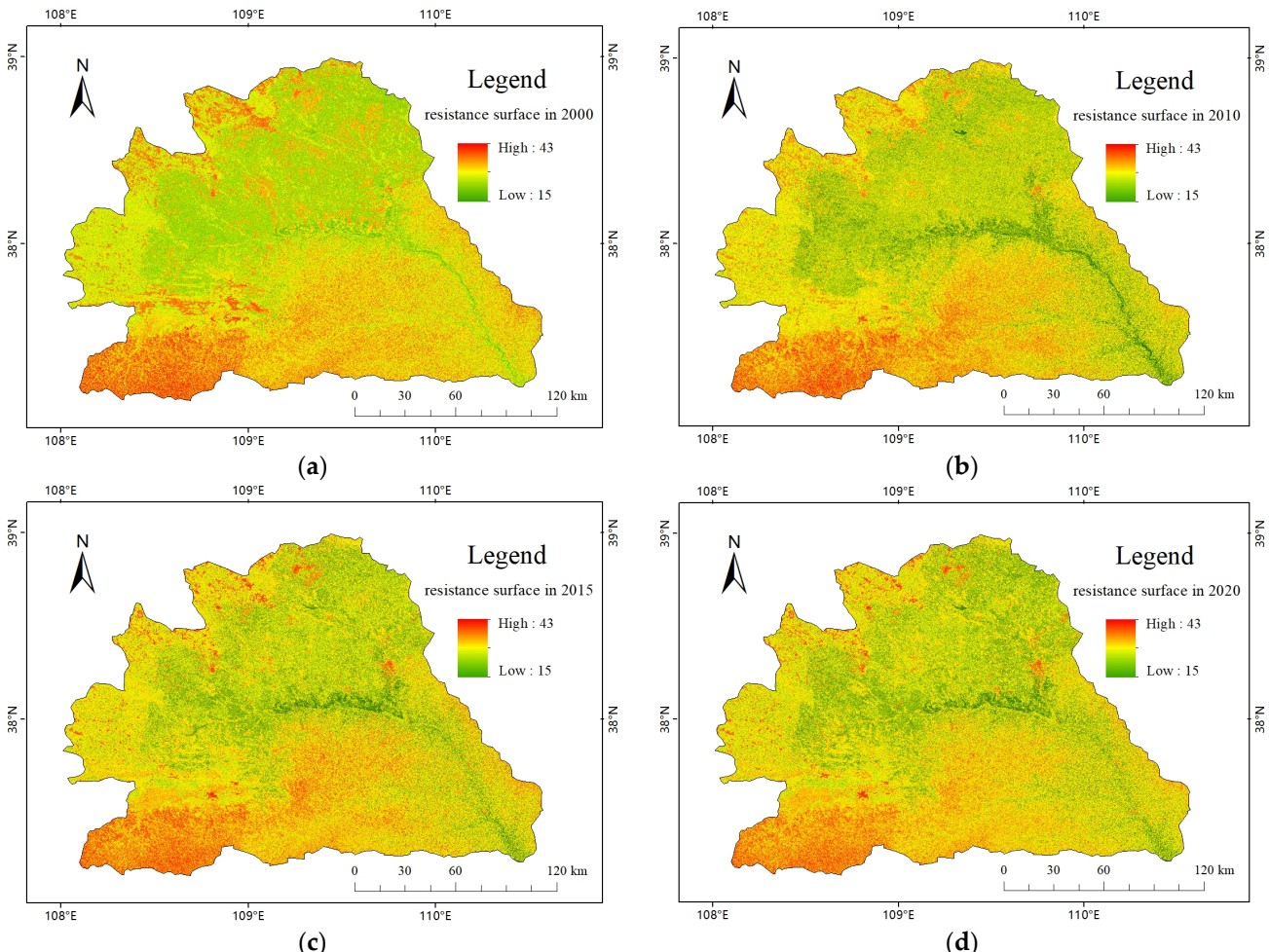

**Figure 8.** Cumulative resistance surface in the study area from 2000 to 2020. (**a**) Cumulative resistance surface in 2000, (**b**) Cumulative resistance surface in 2010, (**c**) Cumulative resistance surface in 2015, (**d**) Cumulative resistance surface in 2020.

According to the cumulative resistance surface model, the potential ecological corridors in the Wuding River Basin summed to 649 in 2000, 451 in 2010, 642 in 2015, and 454 in 2020. The highest number of potential ecological corridors was in 2000, with dense distribution in three areas: the southeast corner, the northeast of the central region, and the southwest direction. The length of individual corridors was relatively short. In the central and southern parts of the basin and the northwest direction of the central region, the distribution of corridors was relatively sparse, and the length of individual corridors was longer. There were no corridors in the northernmost and southwestern regions adjacent to the northwestern boundary of the area.

In 2010, the number of ecological corridors was the lowest, with a relatively uniform distribution and a wider overall coverage range, covering the northern and northwestern areas without corridors in 2000. However, the coverage range in the southeast corner was slightly smaller than in 2000, and the northeast corner of the southeast and central regions of the basin had the densest distribution of corridors, while the distribution of corridors in other areas was relatively uniform.

In 2015, there was a dense distribution of potential ecological corridors in the three areas of the east, west, and center near the southern boundary line of the basin, as well as in the central and eastern regions. Several source areas in the central and southern parts radiated outwards to connect other source areas, with no corridors in the northern and central-western regions, and relatively sparse distribution in other areas. The distribution pattern was significantly different from that of 2010.

In 2020, the distribution range expanded again, with the addition of corridors in the northern and central-western regions. The densely distributed areas were still in the southeast and northeast parts of the basin, and the region with the densest distribution of corridors shifted westward in the loess hill and gully area.

The overall coverage of ecological corridors in 2020 is wider than that of 2000, with larger areas covered in the northern and southwestern corners, thereby widening the north-south width of the network of ecological corridors in the study area. However, in the southeastern corner and strip-shaped areas along the eastern boundary, ecological corridors were not present in 2020, whereas the 2000 forest and grass ecological spatial network included them. Overall, the distribution of the forest and grass ecological spatial network formed by ecological corridors in 2020 is more evenly distributed, with densities at various locations being more similar than in 2000. In 2000, the distribution of ecological corridors in the study area was more uneven, with high density in the eastern part and the southwestern direction of the central part, while other areas were relatively sparse or even without ecological corridor distribution. The distribution of ecological corridors from 2000 to 2020 is shown in Figure 9.

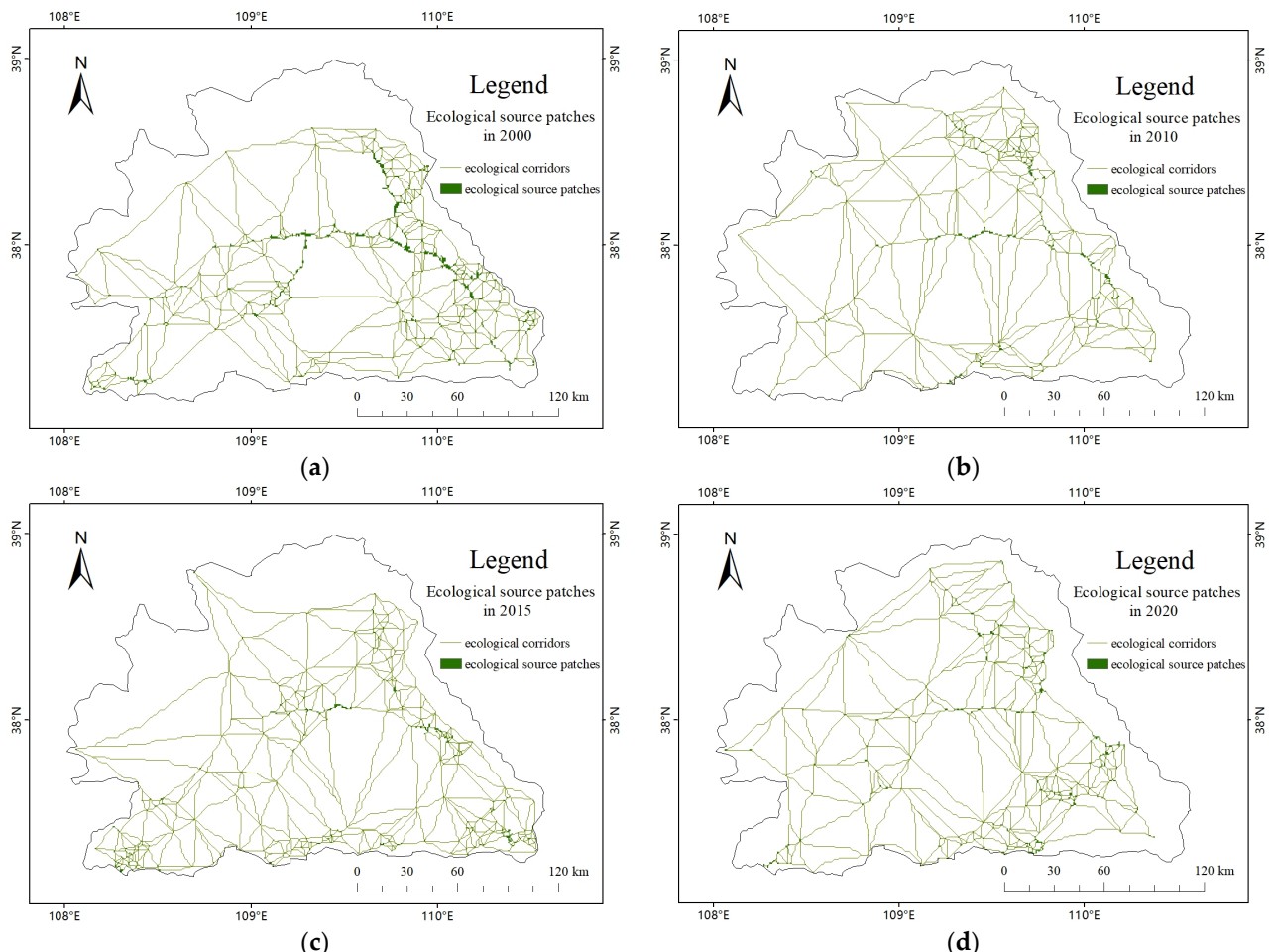

**Figure 9.** Ecological corridors in the study area from 2000 to 2020. (**a**) Ecological corridors in 2000, (**b**) Ecological corridors in 2010, (**c**) Ecological corridors in 2015, (**d**) Ecological corridors in 2020.

### 3.3. Basic Characteristics of Complex Networks

3.3.1. Degree and Degree Distribution

The degree distribution of each year's complex networks within the study area was calculated using Matlab and Python. In 2000, there were 320 ecological nodes within the study area that could be discussed in terms of their degree and degree distribution. This number decreased to 228 in 2010, increased to 284 in 2015, and then decreased again to 207 in 2020. Overall, there was a fluctuating trend in the total number of ecological nodes, with a significant reduction observed in the total number of nodes.

In the year 2000, the range of degrees was between 0 and 9, and there were no ecological nodes with a degree of 0. There were 3 nodes with a degree of 1, 41 nodes with a degree of 2, 82 nodes with a degree of 3, 86 nodes with a degree of 4, 57 nodes with a degree of 5, 29 nodes with a degree of 6, 16 nodes with a degree of 7, and 6 nodes with a degree of 8, for a total of 320 nodes with an average degree of 4.04.

In 2010, the degree range was between 0 and 15, and there were no ecological nodes with a degree of 0. There were 3 nodes with a degree of 1, 32 nodes with a degree of 2, 74 nodes with a degree of 3, 52 nodes with a degree of 4, 33 nodes with a degree of 5, 17 nodes with a degree of 6, 8 nodes with a degree of 7, 6 nodes with a degree of 8, 1 node with a degree of 9, 1 node with a degree of 11, and 1 node with a degree of 14, for a total of 228 nodes with an average degree of 3.96.

In 2015, the degree range was between 0 and 11, and there were no ecological nodes with a degree of 0 or 1. There were 18 nodes with a degree of 2, 64 nodes with a degree of 3, 67 nodes with a degree of 4, 65 nodes with a degree of 5, 45 nodes with a degree of 6, 12 nodes with a degree of 7, 9 nodes with a degree of 8, 3 nodes with a degree of 9, and 1 node with a degree of 10, for a total of 284 nodes with an average degree of 4.52.

In 2020, the degree range was between 0 and 15, and there were no ecological nodes with a degree of 0. There were 3 nodes with a degree of 1, 13 nodes with a degree of 2, 43 nodes with a degree of 3, 68 nodes with a degree of 4, 38 nodes with a degree of 5, 26 nodes with a degree of 6, 7 nodes with a degree of 7, 4 nodes with a degree of 8, 3 nodes with a degree of 9, and 1 node each with a degree of 10 and 14, for a total of 207 nodes with an average degree of 4.39.

For all four periods, over 75.4% of the nodes had a degree of less than or equal to 5, and except for 2015, the proportion was close to or exceeded 80%. This indicates that most ecological nodes within the watershed are adjacent to each other, but the degree of adjacency still needs to be increased. Degree distribution of the forest–grassland ecological network in the study area from 2000 to 2020 is shown in Figure 10.

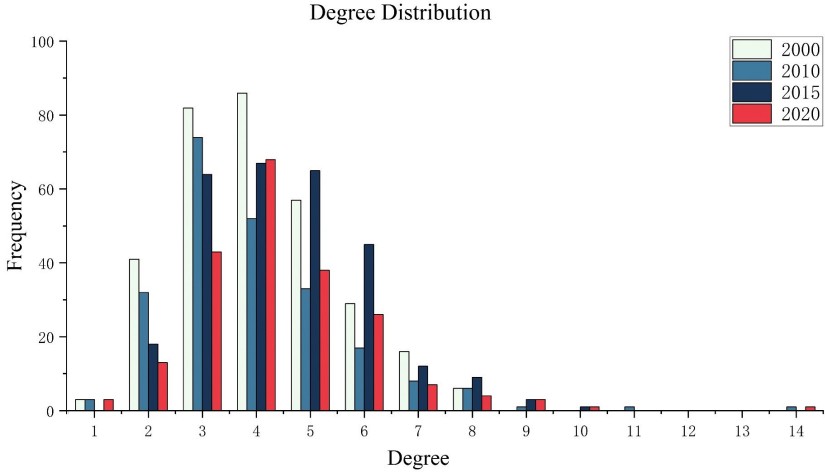

**Figure 10.** Degree distribution of the forest–grassland ecological network in the study area from 2000 to 2020.

### 3.3.2. Average Path Length

The diameter of the potential forest and grassland ecological network in the study area was 8.75 in 2000, with an average path length of 23; the diameter was 6.51 in 2010, with an average path length of 14; the diameter was 7.95 in 2015, with an average path length of 22; and the diameter was 6.47 in 2020, with an average path length of 19. The results indicate that the forest and grassland ecological network in the study area has a small diameter and a relatively long average path length, and it does not exhibit small-world characteristics. There is some blocking of energy flow in the forest and grassland ecological network.

The diameter of the constructed forest and grassland ecological spatial network has not changed significantly, fluctuating between 6.47 and 8.75 from 2000 to 2020. The diameter was the largest in 2000 and the smallest in 2020, indicating relatively compact and convenient energy transfer between ecological nodes. The average path length has a large variation, ranging from 14 to 23, with obvious fluctuations. The highest value was observed in 2000 and the lowest in 2010. However, the overall average path length is relatively long. These results indicate that the forest and grassland ecological network in the study area has a relatively small diameter, a long average path length, and lacks small-world characteristics. Energy flow within the forest and grassland ecological network may be obstructed to a certain extent. Diameter and average path length of forest–grassland ecological network in the study area from 2000 to 2020 is shown in Figure 11.

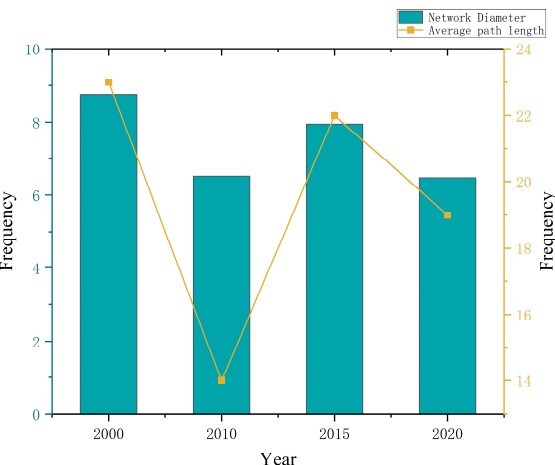

**Figure 11.** Diameter and average path length of forest–grassland ecological network in the study area from 2000 to 2020.

### 3.3.3. Clustering Coefficient

The clustering coefficients of the potential forest and grassland ecological network is shown in Figure 12, and their values reflect the degree of clustering of ecological nodes, usually ranging from 0 to 1. In 2000, there were 69 nodes with a clustering coefficient of 0, 29 nodes with a coefficient of 0.1, 54 nodes with a coefficient of 0.2, 93 nodes (the most nodes) with a coefficient of 0.3, 21 nodes with a coefficient of 0.4, 30 nodes with a coefficient of 0.5, 21 nodes with a coefficient of 0.6, and 4 nodes with a coefficient of 0.9. In 2010, there were 57 nodes with a clustering coefficient of 0, 22 nodes with a coefficient of 0.1, 41 nodes with a coefficient of 0.2, 68 nodes (the most nodes) with a coefficient of 0.3, 9 nodes with a coefficient of 0.4, 16 nodes with a coefficient of 0.5, 12 nodes with a coefficient of 0.6, and 3 nodes with a coefficient of 0.9. In 2015, there were 30 nodes with a clustering coefficient of 0, 19 nodes with a coefficient of 0.1, 57 nodes with a coefficient of 0.2, 73 nodes (the most nodes) with a coefficient of 0.3, 35 nodes with a coefficient of 0.4, 41 nodes with a coefficient of 0.5, 27 nodes with a coefficient of 0.6, and 2 nodes with a coefficient of 0.9. In 2020, there were 29 nodes with a clustering coefficient of 0, 18 nodes with a coefficient of 0.1, 34 nodes with a coefficient of 0.2, 60 nodes (the most nodes) with a coefficient of 0.3, 22 nodes with a

coefficient of 0.4, 24 nodes with a coefficient of 0.5, 16 nodes with a coefficient of 0.6, and 4 nodes with a coefficient of 0.9.

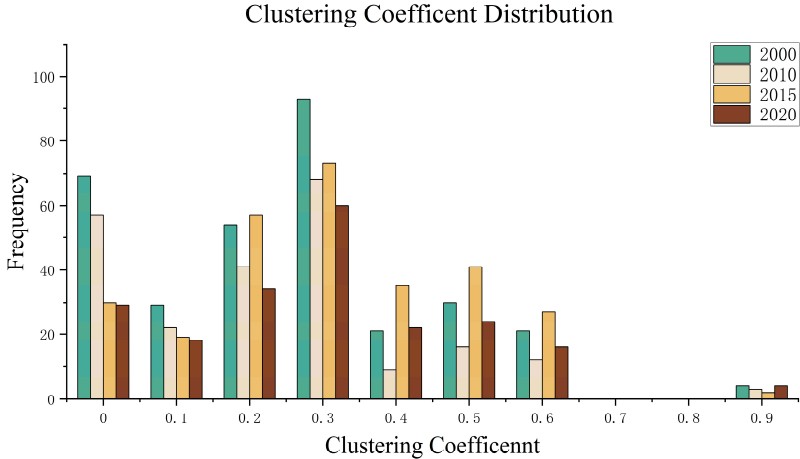

**Figure 12.** Clustering coefficients of nodes of forest–grassland ecological network in the study area from 2000 to 2020.

From 2000 to 2020, the number of ecological nodes with a clustering coefficient of 0.3 was consistently the highest, while the number of ecological nodes with a clustering coefficient of 0.9 was consistently the lowest. Furthermore, there were no ecological nodes with clustering coefficients of 0.7 or 0.8 in any of the four periods. Nodes with a clustering coefficient of 0 have no clustering characteristics, and there is no connectivity between them and the surrounding nodes, making it impossible for energy to flow. Most of the clustering coefficients of ecological nodes in the potential forest and grassland ecological network in the basin are below 0.6. Combined with the average path length of the forest and grassland ecological network, it can be seen that the clustering degree of the nodes in the basin is low, and the connectivity of the nodes is poor.

### 3.4. Exploration of Correlations

Based on the theory of complex networks, this study used Origin and Gephi software to analyze the correlation between the topological indicators and ecological service functions of the forest and grass ecological spatial network using the method of Pearson correlation analysis. The 2020 data were chosen as a proxy for the correlation study, as they are newer, closer to the current situation, and more informative and meaningful for research than the previous three periods. Using 2020 data as an example, the correlations between numerous topological indicators and three ecological service function indicators (windbreak and sand fixation, soil conservation, and carbon sequestration) are explored. The windbreak and sand fixation service capacity index is positively correlated with degree, clustering, triangles, and eigencentrality, and negatively correlated with eccentricity, betweenness centrality, bridging coefficient, authority, hub, and cluster. The correlation with cluster is the strongest, with 99% certainty, and the correlation with eccentricity, bridging centrality, triangles, and eigencentrality is relatively strong. The soil conservation amount is positively correlated with eccentricity, bridging coefficient, bridging centrality, and cluster, and negatively correlated with degree, closeness centrality, authority, hub, clustering, triangles, and eigencentrality. The correlation with eigencentrality has the highest certainty, reaching 99%, and cluster, clustering, and triangles also have a 95% certainty. NPP is negatively correlated with degree, eccentricity, closeness centrality, harmonic closeness centrality, betweenness centrality, authority, hub, cluster, and eigencentrality, and the overall correlation is not significant. The correlations with eccentricity and eigencentrality are relatively strong. The correlation between the windbreak and sand fixation service capacity index, soil conservation quantity, NPP and complex network topology in 2020 is shown in Figure 13.

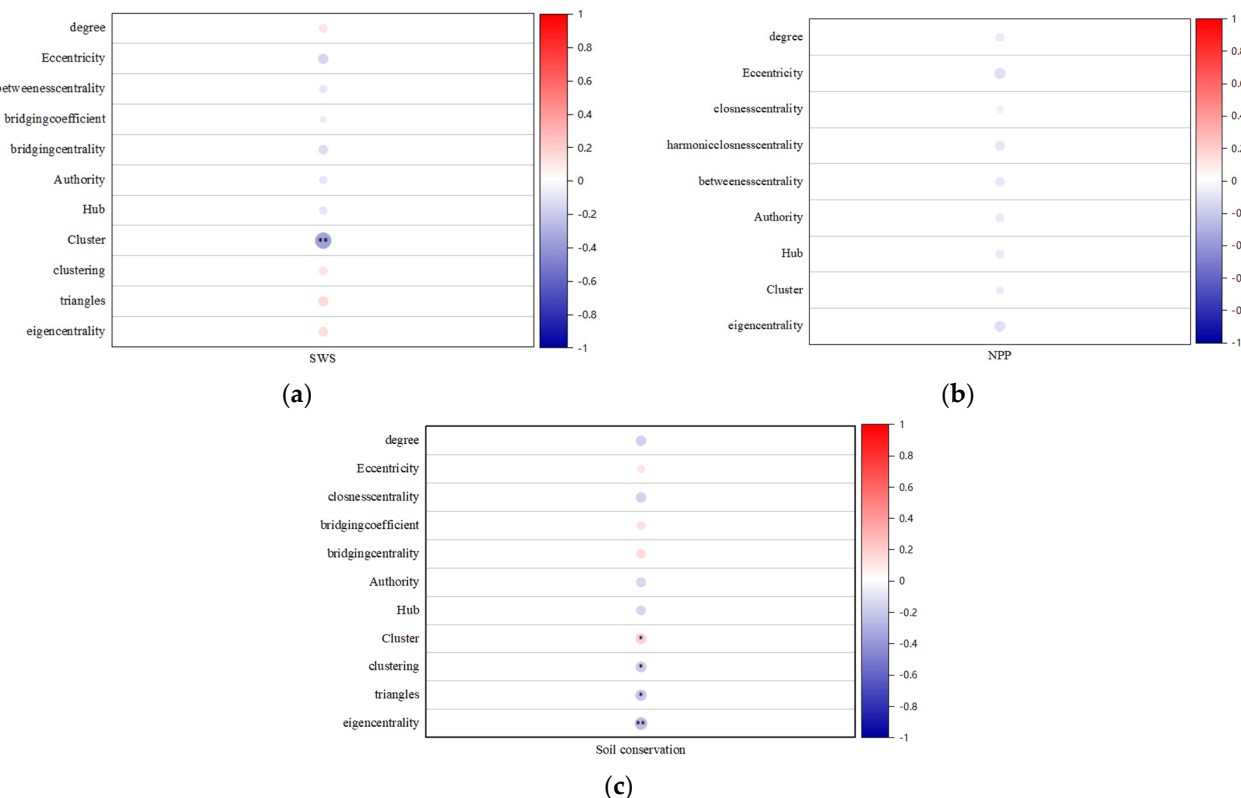

**Figure 13.** The correlation between the windbreak and sand fixation service capacity index (**a**), soil conservation quantity (**b**), NPP (**c**) and complex network topology in 2020. (* $p \leq 0.05$; ** $p \leq 0.01$).

## 4. Discussions

### 4.1. Ecological Network Construction

The article differs from many scholars who select areas with specific land cover types such as forests, shrubs, grasslands, and water bodies as ecological source areas [10,25,58–60] by selecting regions with high values of RSEI, which to some extent provides a more objective and comprehensive evaluation of the ecological environment. The formation of ecosystems and ecological spaces is influenced by various factors, not just a single ecological factor. RSEI calculation is entirely based on remote sensing information and covers and reflects multiple intuitive and different aspects of ecological environment indicators [31,61], which compensates for the subjective weighting of various ecological factors in early research, enabling rapid monitoring and evaluation of regional ecological environments with a high reference value. This model has been widely used in urban evaluation, but there are fewer studies on watersheds [62]. This paper is a more innovative application of the indicator model to the study of watersheds. After identifying regions with high RSEI values as ecological source areas, a cumulative minimum resistance surface was constructed based on various ecological factors, followed by the extraction of ecological corridors using the MCR model to build a forest and grass ecological spatial network in the study area. This construction method and process have been widely recognized and applied with proven reliability in previous research [10,63–65].

In landscape ecology, different source areas have different weights due to their location, patch size, shape, and other factors, and it is difficult to evaluate ecological flow between ecological source areas concretely. Therefore, it is intuitive and necessary to quantify the forest and grass ecological space network in conjunction with complex network theory to better understand and evaluate it. This article is an innovative approach to simultaneously study the spatial network of forest and grass ecology and ecological service functions in the study area in a spatial and temporal sequence, and explore the correlation between the two, which can provide targeted guidance for ecological restoration and conservation.

### 4.2. Ecological Network Evaluation and Optimization Suggestions

The forest and grass ecological spatial network density in the Wuding River Basin exhibits obvious spatial heterogeneity, with an overall pattern of denser southeastern and sparser northwestern areas, with decreasing density from southeast to northwest. The ecological source areas in the study area are generally fragmented, especially in the northwest, where there are few ecological source areas and severe fragmentation. This may be due to the difference in altitude between the southeastern and northwestern sides, which creates obstacles to the transfer of ecological energy to some extent. At the same time, the northwestern part of the study area belongs to the sandy area, where the surface is mostly covered by sand dunes, and vegetation is scarce. In contrast, the southeastern part belongs to the loess area, with relatively abundant precipitation, which is conducive to vegetation growth. Therefore, a transition from a semi-arid to arid region is formed from southeast to northwest, as well as a transition from loess to sand, which has some influence on the distribution of ecological source areas, ecological nodes, and ecological corridors.

In recent years, the rapid increase in population and excessive grazing have led to the overall destruction of surface vegetation in the northwest region of China, resulting in soil desertification and sandification [66]. To improve the ecological environment, measures can be taken to advocate the use of clean energy (such as wind and solar energy), prohibit grazing in degraded grassland and ecologically fragile areas, restrict and manage excessive grazing through national policies, and reduce deforestation and destruction of vegetation. Targeted ecological restoration work can also be carried out by planting shrubs and grasses, planting windbreak forests, and performing other operations to restore vegetation, improving the ecological environment from both the protection and restoration perspectives.

### 4.3. Correlation Conclusion Discussion

This article explores the inherent relationship between the topological indicators of forest and grassland ecological spatial networks and ecosystem service functions and conducts correlation analysis on various topological indicators and three important ecosystem service functions. The results show that the windbreak and sand fixation service capacity index and cluster have a significant negative correlation. The soil retention and relatively more topological indicators have a correlation probability of over 95%, among which the correlation with the centrality features is more significant and negative. NPP is negatively correlated with the studied topological indicators, but there is no clear understanding. Wang [67] et al.'s research results show that the spatial distribution of ecological nodes based on topological indicators is similar to that based on NPP, and the topological features of ecological nodes in specific land cover types are related to their carbon sequestration capacity. Qiu [67] et al.'s research results show that carbon sequestration capacity and betweenness centrality show a significant negative correlation, consistent with the results of this study.

### 4.4. Research Limitations and Future Research Directions

The present study has several limitations, which could be improved in the future: (1) When analyzing the topological indicators of the forest and grassland ecological spatial network for each period, only a few indicators such as degree and degree distribution, diameter, average path length, and clustering coefficient were considered. In the future, more topological indicators could be introduced for detailed analysis. (2) This study only utilized three indicators—windbreak and sand fixation, soil conservation, and carbon fixation—to research and analyze ecosystem services. It is possible to incorporate additional indicators such as carbon–water use efficiency [68,69] and climate regulation [70,71] for evaluation. (3) This study only constructed and analyzed resistance surfaces, and further consideration could be given to factors such as resilience [72].

## 5. Conclusions

By using the RSEI to extract ecological source areas and combining five main ecological factors, including elevation, slope, NDVI, NDWI, and land use data, an ecological resistance surface was constructed. Using the MCR model, the potential ecological corridors in the study area were extracted. The results show the following:

(1) Spatiotemporal variations in ecological service functions: From 2000 to 2020, the ecological service functions in the study area have all been improved. The overall windbreak and sand fixation ability has been improved. The ecological service functions in the study area have generally improved, with the average fixed sand amount increasing by 2.7%, the average NPP increasing by 95.7%, and the average soil retention increasing by 912.70 tons, an increase of 16.17%.

(2) Temporal and spatial changes in the forest and grassland ecological spatial network: The overall ecological flow in the study area is good, and the potential ecological corridors cover almost the entire watershed, with an increase in the area covered by the corridor network. However, overall, the ecological nodes are relatively fragmented, with a low clustering degree and poor connectivity. The average length of ecological corridors is long, and their distribution is scattered, which may lead to ecological flow stagnation and singularity.

(3) Correlation between topological indicators and ecological service functions: The windbreak and sand fixation service capacity index in the study area showed a very strong negative correlation with cluster, with a confidence level of 99%. Moreover, it exhibited a strong negative correlation with eccentricity and bridging centrality, and a strong positive correlation with triangles and eigencentrality. Soil conservation was found to have a very strong negative correlation with eigencentrality, with a maximum confidence level of 99%. Additionally, it showed a strong positive correlation with cluster at a confidence level of 95% or higher, and a strong negative correlation with clustering and triangles, with an error probability of less than 5%. NPP was negatively correlated with the verified topological indicators, but the overall correlation was not significant. Among them, it showed a relatively high correlation with eccentricity and eigencentrality.

This article fills a gap in the research on ecological spatial networks and ecosystem service functions in the Wuding River Basin. It investigates the spatio-temporal changes of the forest and grass ecological spatial network and ecosystem service function in the study area and explores the relationship between these two aspects. This can provide reference suggestions for the ecological environment protection in this region.

**Supplementary Materials:** The following supporting information can be downloaded at: https://www.mdpi.com/article/10.3390/rs15092456/s1, Figure S1: Ecological resistance factors of the study area in the year 2000. (a) DEM in 2000, (b) Slope in 2000, (c) NDVI in 2000, (d) NDWI in 2000, (e) Land use type in 2000; Figure S2: Ecological resistance factors of the study area in the year 2010. (a) DEM in 2010, (b) Slope in 2010, (c) NDVI in 2010, (d) NDWI in 2010, (e) Land use type in 2010; Figure S3: Ecological resistance factors of the study area in the year 2015. (a) DEM in 2015, (b) Slope in 2015, (c) NDVI in 2015, (d) NDWI in 2015, (e) Land use type in 2015; Figure S4: Windbreak and Sand Fixation in the Study Area from 2000 to 2020. (a) Windbreak and Sand Fixation in 2000, (b) Windbreak and Sand Fixation in 2010, (c) Windbreak and Sand Fixation in 2015, (d) Windbreak and Sand Fixation in 2020; Figure S5: Soil Conservation Amount of the Study Area from 2000 to 2020. (a) Soil Conservation Amount in 2000, (b) Soil Conservation Amount in 2010, (c) Soil Conservation Amount in 2015, (d) Soil Conservation Amount in 2020; Figure S6: NPP of the Study Area from 2000 to 2020. (a) NPP in 2000, (b) NPP in 2010, (c) NPP in 2015, (d) NPP in 2020.

**Author Contributions:** Y.Z. performed data treatments and wrote the paper; X.W., J.M., C.X., S.Q., W.L. and F.W. contributed to the discussion of the results; Q.Y. contributed some ideas and revised the paper; and all authors edited the paper. All authors have read and agreed to the published version of the manuscript.

**Funding:** This work was supported by the National Natural Science Foundation of China (NSFC) project "Research on the Method of Delineating Ecological Protection Red Line in Desert Oasis Areas Based on Ecological Security Patterns" (420071237) and the National Key R&D Program of China [2022YFE0127700].

**Data Availability Statement:** Landsat, DEM, and meteorological data used in this study are publicly available in public repositories. Landsat data is accessible from Google Earth Engine (http://code. earthengine.google.com/, accessed on 8 December 2022). The DEM data for the study area were obtained from the GDEMV3 digital elevation data product of the Geographic Spatial Data Cloud (http://www.gscloud.cn/, accessed on 14 December 2022). Monthly average temperature and precipitation data can be obtained from Resource and Environment Science and Data Center of the Chinese Academy of Sciences (http://www.resdc.cn/, accessed on 20 December 2022).

**Acknowledgments:** We would like to thank Google Earth Engine (http://code.earthengine.google. com/, accessed on 8 December 2022) for providing the remote sensing images of Landsat 5 and Landsat 8. We also appreciate the support of NASA for providing the MODIS Gross and Net Primary Productivity (MOD17A2H). We would also like to acknowledge the valuable suggestions from the anonymous reviewers on this manuscript.

**Conflicts of Interest:** The authors declare no conflict of interest.

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
