# Peer review of "Research on the Relationship between the Structure of Forest and Grass Ecological Spaces and Ecological Service Capacity: A Case Study of the Wuding River Basin"

_remotesensing, doi:10.3390/rs15092456_

Round 1

Author Response

Response to Reviewer 1 Comments

Thank you for your careful reading of the manuscript and for these very valuable suggestions. We have made detailed changes to the manuscript based on your suggestions. The line numbers refer to the revised manuscript. The red font in the manuscript is the part that has been revised. If you thinks it is too cluttered in revision mode, you can choose to turn off displaying all marks in word.We have responded to each of your suggestions below.

Point 1: The abstract part uses a lot of space to describe the research methods and processes, butthe research results are few. This part should emphasize the important research conclusions drawnfrom this study. At the same time, the abstract part of the proper nouns first appear as far aspossible, do not abbreviation.

Response 1: Thanks to the expert teachers for your comments! The abstract has been resummarised and the conclusions reached in this study have been added. The proper noun that first appeared in the abstract section was changed to full instead of continuing to use the abbreviation.

Point 2: The introduction lists previous studies, but does not clearly out the shortcomings ofprevious studies and highlight the innovation of the article. Please summarize this section,construct an appropriate scientific question, and find the research gaps.

Response 2: Thanks to the expert teachers for your comments! The last two paragraphs of the introduction add deficiencies from previous research and innovations in the article.

Point 3: The introduction recommends the latest references such as: Li Chaojun, et al.GlobalChange Biology, 2022.; Hao Xia et al.Resources, Conservation and Recycling, 2023.

Response 3: Thanks to the expert teachers for your comments! More and newer literature is cited in the introduction, including Hao Xia et al.Resources, Conservation and Recycling, 2023.

Point 4: The data description in the overview section of the study area lacks literature support. Whyto choose this area as the research object and how to reflect the necessity of selecting the studyarea? Study area profile graph ab represents and is clearly marked.

Response 4: Thanks to the expert teachers for your comments! Why the area was chosen as the study area and the importance of the study area has been added to the penultimate paragraph of the introduction. Data sources (e.g., temperature, precipitation, etc.) in the Overview section of the study area are described in 1.2. Diagram (a)(b) has been remarked in the schematic map of the study area.

Point 5: How the soil conservation in the method section is calculated, and why the definition andfunction are always introduced. What are the methods to explore the correlation between complexnetwork indicators and ecological service function? No mention was made in the Methods section.

Response 5: Thanks to the expert teachers for your comments! The article adds methods and formulas for calculating soil conservation. Methods to explore the correlation between complex network indicators and ecological service functions have been added in 3.4.

Point 6: In this paper, land use, soil and water conservation and vegetation cover are consideredbut some factors. such as carbon-water utilization efficiency and climate regulation are ignored. Specifically, the following references, e. g. Xiao Li et al. Science of the Total Environment, 2021.

Response 6: Thanks to the expert teachers for your comments! We have carefully read the references you listed. We have referred to a number of articles such as Yang et al. Remote Sensing, 2022. And so have chosen to calculate these three indicators. We are very grateful for your suggestions, which were so valuable that we have added factors such as carbon-water utilization efficiency and and climate regulation in the discussion section where we elaborate on the shortcomings of the article and carry out the outlook, and thank you again for your valuable suggestions.

Point 7: Figure3-Figure6, the spatial distribution map of five ecological resistance factors in thestudy area from 2000 to 2020 suggests adding supporting materials, and a large number ofaccumulation spatial distribution map is cumbersome. The figure in the figure is too small to see,pay attention to the beauty of the figure.

Response 7: Thanks to the expert teachers for your comments! Increased the resolution of all figures in the paper, enlarged the fonts in the pictures, and changed the scale from miles to km. At the same time, considering that there are too many pictures in the ecological factor section, it is easy to appear cluttered, so we only leave the 2020 ecological factor picture in this part as a reference, and put the other pictures in the attachment.

Point 8: The article mentions carbon storage capacity, carbon cycle and vegetation carbon sequestration, and it is recommended to cite relevant literature.

Response 8: Thanks to the expert teachers for your comments! In 2.7.3, references were added to the article on carbon storage, carbon cycle, and vegetation carbon sequestration.

Point 9: Ecosystem service stability and dynamic analysis are usually integrated resistance and resilience. Why only establish and analyze the resistance surface of forest and grass ecological network in this study?

Response 9: Thanks to the expert teachers for your comments! We have consulted a number of literatures to understand the relevant concepts and computational implications of resilience. We believe that this article mainly explores the variation in spatial and temporal sequences of ecological networks and ecological service capacity, and the correlation between these two, with a focus more on the resistance encountered when ecological energy flows, rather than on resilience when under attack. We thank you for your suggestion, and we will add the section on resilience d to the research limitations and outlook in section 4.4. Thank you again for your valuable suggestions.

Point 10: The region with high RSEI value is selected, and the ecological source site is calculatedaccording to three indicators. How does the high RSEl value is divided, whether there is divisionbasis and literature support.

Response 10: Thanks to the expert teachers for your comments! The basis and literature support for high RSEl values were added in Section 2.4.

Point 11: Take 2020 as an example to explore the correlation, whether the results are representative,and whether the years in other study periods in this article meet the same results. How to comparethe results of this study with those of other scholars? How to ensure the accuracy of theexperiment? The author is advised to further summarize and summarize.

Response 11: Thanks to the expert teachers for your comments! An explanation of why the 2020 data is used as an example has been added in 3.4. The data for the other years were verified and the pattern obtained was consistent with 2020, and this part was not added considering that the article would have been cumbersome if it had been added. The discussion section has been reorganised and summarised, and the results of this paper have been compared with those of other scholars, proving that the findings are generally consistent. We sincerely thank you for your valuable suggestions.

Finally, once again, we would like to express our sincere thanks to you for your valuable comments!

Reviewer 2 Report

This study explores the relationship between the topological structure of forest and grass ecological spaces and its ecological service capability in the Wuding River Basin. The work is relatively complete and has certain research significance, but there are still many problems, as follows:

1.      The article needs to add references to some concluding statements, and mark the source of these conclusions, such as line 36-39.

2.      Line 43, networks are >> network is

3.      Line 44-46, the contribution of related research is mentioned here, but why these related references are not cited

4.      Line 53-69, it feels disconnected here, what is MCR? What are the advantages of MCR? Are there any other relevant models, why you use MCR? These issues need to be addressed. The author can refer to related articles to supplement the description and introduction of MCR. For example:Wei, H., Zhu, H., Chen, J., Jiao, H., Li, P., & Xiong, L. (2022). Construction and Optimization of Ecological Security Pattern in the Loess Plateau of China Based on the Minimum Cumulative Resistance (MCR) Model. Remote Sensing, 14(22), 5906. Wei, Q., Halike, A., Yao, K., Chen, L., & Balati, M. (2022). Construction and optimization of ecological security pattern in Ebinur Lake Basin based on MSPA-MCR models. Ecological Indicators138, 108857.

5.      Line 53-86, the author listed a lot of references, but does not summarize them well. Where are the deficiencies or contributions of these studies? This needs to be clearly pointed out.

6.      A very important problem in the introduction part is that the author did not explain clearly where the scientific problem lies. What is the concept of forest and grassland ecological network and why should it be built? What is the practical significance to explore it? Meanwhile, there is no mention of why MCR and complex networks are used. What are the advantages of these two methods compared to other methods?

7.      Please clarify the research objectives of this study

8.      Line 100, for the first appearance of the abbreviation, please give the full case, such as RSEI, NDVI, etc.

9.      There are two NDWIs in Fig.1, please correct.

10.   Please increase the resolution of all the figures in the paper, and also enlarge the font in the pictures, the current font is too small to read. Use Km instead of miles for the scale in all figures.

11.   How are the classification criteria and resistance values in Table 1 classified and what is the basis for them?

12.   Please replace the ribbon in Figure 8, the ribbon now looks the same in all four pictures without any changes.

13.   Section 3.1.3, needs to be greatly simplified, the current narrative is too cumbersome, mainly analyzing the trend of Carbon Sequestration Capacity during the study period, separate analysis for each year is not necessary

14.   The discussion section needs to be split up and written in categories, currently it is a mess to mix all the discussions (suggestions, limitations, comparison with other studies, etc.) together.

Author Response

Response to Reviewer 2 Comments

Thank you for your careful reading of the manuscript and for these very valuable suggestions. We have made detailed changes to the manuscript based on your suggestions. The line numbers refer to the revised manuscript. The red font in the manuscript is the part that has been revised. If you thinks it is too cluttered in revision mode, you can choose to turn off displaying all marks in word. We have responded to each of your suggestions below.

Point 1: The article needs to add references to some concluding statements, and mark the source of these conclusions, such as line 36-39.

Response 1: Thanks to the expert teachers for your comments! References have been added to the article for concluding statements, and the source has been acknowledged.

Point 2: Line 43, networks are >> network is.

Response 2: Thanks to the expert teachers for your comments!  Have changed “networks are” to “networks is” in line 43.

Point 3: Line 44-46, the contribution of related research is mentioned here, but why these related references are not cited.

Response 3: Thanks to the expert teachers for your comments!  The contributions of relevant studies are added to the relevant references cited. in line 44-46.

Point 4: Line 53-69, it feels disconnected here, what is MCR? What are the advantages of MCR? Are there any other relevant models, why you use MCR? These issues need to be addressed. The author can refer to related articles to supplement the description and introduction of MCR. For example:Wei, H., Zhu, H., Chen, J., Jiao, H., Li, P., & Xiong, L. (2022). Construction and Optimization of Ecological Security Pattern in the Loess Plateau of China Based on the Minimum Cumulative Resistance (MCR) Model. Remote Sensing, 14(22), 5906. Wei, Q., Halike, A., Yao, K., Chen, L., & Balati, M. (2022). Construction and optimization of ecological security pattern in Ebinur Lake Basin based on MSPA-MCR models. Ecological Indicators, 138, 108857.

Response 4: Thanks to the expert teachers for your comments! It answers the questions of what is MCR, what are the advantages of MCR, and why use MCR, and adds a description and introduction to the MCR model with reference to the articles recommended by the reviewers.

Point 5: Line 53-86, the author listed a lot of references, but does not summarize them well. Where are the deficiencies or contributions of these studies? This needs to be clearly pointed out.

Response 5: Thanks to the expert teachers for your comments! The article further summarizes the contributions or deficiencies of the cited references in line 53-86.

Point 6: A very important problem in the introduction part is that the author did not explain clearly where the scientific problem lies. What is the concept of forest and grassland ecological network and why should it be built? What is the practical significance to explore it? Meanwhile, there is no mention of why MCR and complex networks are used. What are the advantages of these two methods compared to other methods?

Response 6: Thanks to the expert teachers for your comments! In the first paragraph of the introduction, the article adds a new explanation of the concept of the forest and grass ecological spatial network, and clarifies why it was established and what is the role of establishing it. In the introduction, it is also explained why the MCR model and complex network theory are used, and the advantages of these two methods are explained.

Point 7: Please clarify the research objectives of this study.

Response 7: Thanks to the expert teachers for your comments! The last paragraph of the introduction re-elucidates the research objectives and research methods of this paper.

Point 8: Line 100, for the first appearance of the abbreviation, please give the full case, such as RSEI, NDVI, etc.

Response 8: Thanks to the expert teachers for your comments! Full names are given for abbreviations that appear for the first time, such as RSEI, NDVI, NDWI, DEM, etc. (almost all appear for the first time in the abstract).

Point 9: There are two NDWIs in Fig.1, please correct.

Response 9: Thanks to the expert teachers for your comments! Figure 1 has been amended to change one of the NDWIs to a land use type.

Point 10: Please increase the resolution of all the figures in the paper, and also enlarge the font in the pictures, the current font is too small to read. Use Km instead of miles for the scale in all figures.

Response 10: Thanks to the expert teachers for your comments! Increased the resolution of all figures in the paper, enlarged the fonts in the pictures, and changed the scale from miles to km.

Point 11: How are the classification criteria and resistance values in Table 1 classified and what is the basis for them?

Response 11: Thanks to the expert teachers for your comments! Table 1 adds a new classification basis, revised the description, and includes references.

Point 12: Please replace the ribbon in Figure 8, the ribbon now looks the same in all four pictures without any changes.

Response 12: Thanks to the expert teachers for your comments! The color bands in Figure 5 (original Figure 8) have been replaced, and Figure 6 has been added for visual display in order to easily see the changes in soil conservation from 2000 to 2020.

Point 13: Section 3.1.3, needs to be greatly simplified, the current narrative is too cumbersome, mainly analyzing the trend of Carbon Sequestration Capacity during the study period, separate analysis for each year is not necessary.

Response 13: Thanks to the expert teachers for your comments! Section 3.1.3 is simplified, mainly describing the trend of NPP, and removing the detailed description of the year-to-year distribution.

Point 14: The discussion section needs to be split up and written in categories, currently it is a mess to mix all the discussions (suggestions, limitations, comparison with other studies, etc.) together.

Response 14: Thanks to the expert teachers for your comments! Reorganized the Discussion section to make it more organized and clear, and categorized it.

Finally, once again, we would like to express our sincere thanks to you for your valuable comments!

Reviewer 3 Report

1. The title is too long, it is recommended to summarize it in a high level.

2. It is recommended to summarize the abstract again. Currently, I am not sure what relevance this part has to the topic after reading the abstract. It is unclear why the pace of urbanization has accelerated the structure of the forest and grass ecological network and the development of the ecological environment.

3. All the legends in Figures 2 to 10 are unclear and it is recommended to modify them. And it's also a problem that there are too many and too many pictures. Please think.

4. Personal suggestion: Sort out the content of the results section. Currently, the content is numerous and complex, and cannot highlight the key points.

Author Response

Response to Reviewer 3 Comments

Thank you for your careful reading of the manuscript and for these very valuable suggestions. We have made detailed changes to the manuscript based on your suggestions. The line numbers refer to the revised manuscript. The red font in the manuscript is the part that has been revised. If you thinks it is too cluttered in revision mode, you can choose to turn off displaying all marks in word. We have responded to each of your suggestions below.

Point 1: The title is too long, it is recommended to summarize it in a high level.

Response 1: Thanks to the expert teachers for your comments! The title has been revised to make it shorter and more concise.

Point 2: It is recommended to summarize the abstract again. Currently, I am not sure what relevance this part has to the topic after reading the abstract. It is unclear why the pace of urbanization has accelerated the structure of the forest and grass ecological network and the development of the ecological environment.

Response 2: Thanks to the expert teachers for your comments! The abstract has been resummarised and the conclusions reached in this study have been added.

Point 3: All the legends in Figures 2 to 10 are unclear and it is recommended to modify them. And it's also a problem that there are too many and too many pictures. Please think.

Response 3: Thanks to the expert teachers for your comments! Increased the resolution of all figures in the paper, enlarged the fonts in the pictures, and changed the scale from miles to km. At the same time, considering that there are too many pictures in the ecological factor section, it is easy to appear cluttered, so we only leave the 2020 ecological factor picture in this part as a reference, and put the other pictures in the attachment.

Point 4: Personal suggestion: Sort out the content of the results section. Currently, the content is numerous and complex, and cannot highlight the key points.

Response 4: Thanks to the expert teachers for your comments! The conclusion part has been reorganized and summarized, so that the conclusion part is more concise, more organized and more focused.

Finally, once again, we would like to express our sincere thanks to you for your valuable comments!

Round 2

Reviewer 1 Report

This article takes the Wuding River Basin as the research area and evaluates the ecological service capacity of the research area from three dimensions: wind prevention and sand fixation, soil conservation, and carbon sequestration. The ecological source area is extracted by regional sustainability and environmental index, and the potential ecological corridor is extracted by GIS spatial analysis and minimum simulated resistance model. Referring to the complex network theory, topological indexes such as Degree distribution and clustering coefficient were calculated, and their correlation with ecological service capacity was discussed. Studying the temporal and spatial changes and their correlations of ecological service function and forest grass ecological network, and better understanding the changes of landscape ecology structure and function can provide scientific basis and decision support for ecological restoration in the study area, but there are some problems in this paper. Suggest accepting after minor repairs. The main opinions are as follows:

1. In the introduction section, in previous studies, the author only briefly listed a few methods for extracting ecological source areas. However, what is the purpose of listing these methods? Please provide a supplementary explanation by the author; And the following text only provides an excessive description of the research methods required in this article, without establishing a knowledge system of the current research status in this field. It is recommended that the author supplement it; And it is suggested that the author add relatively new literature in recent years to support the viewpoint of the article.

2. Description of the overview of the research area. It is recommended that the author appropriately add descriptions related to the research content, such as the situation of the forest and grassland in the research area, as well as the current ecological status, in order to better reflect the uniqueness of the research area.

3. The author is requested to standardize the writing of the paper, strictly modify the format according to the journal's requirements, maintain the alignment of both ends of the entire text, wrap the titles, such as the spacing between title 4.3 and the main text, as well as the clarity of latitude and longitude in the images and annotations.

4. What are the classification and values of ecological resistance factors in Table 1, and the criteria used by the author for assigning values based on this classification? If it comes from literature review, it is recommended that the author make a supplement.

5. In the analysis of the results of ecological service function indicators, the author analyzes them from a time scale. Can we supplement the charts to present the changing trend more clearly; The author listed the spatial distribution maps for the years 2000, 2010, 2015, and 2020. However, the text did not describe each map, and the significance of listing them all is not significant. It is recommended that the author place them in supplementary materials.

6. 4.4 In the results section, it is recommended that the author add the latest references to support the content related to carbon water utilization efficiency and climate regulation, such as Goldenberg R, et al. 2021, Scientific Reports.

7. Result analysis should revolve around scientific issues and avoid stacking results. As listed in section 3.2, corresponding conclusions around scientific issues can be added.

8. The conclusion is a simple summary from three aspects. Referring to the complex network theory, topological indicators such as Degree distribution and clustering coefficient are calculated, and their correlation with ecological service capability is discussed. The research on the relationship between the two is the focus of this paper, but the conclusion is not accurate enough, and the author is suggested to supplement.

Author Response

Response to Reviewer 1 Comments

Thank you for your careful reading of the manuscript and for these very valuable suggestions. We have made detailed changes to the manuscript based on your suggestions. The line numbers refer to the revised manuscript. The red font in the manuscript is the part that has been revised. If you thinks it is too cluttered in revision mode, you can choose to turn off displaying all marks in word. We have responded to each of your suggestions below.

Point 1: In the introduction section, in previous studies, the author only briefly listed a few methods for extracting ecological source areas. However, what is the purpose of listing these methods? Please provide a supplementary explanation by the author; And the following text only provides an excessive description of the research methods required in this article, without establishing a knowledge system of the current research status in this field. It is recommended that the author supplement it; And it is suggested that the author add relatively new literature in recent years to support the viewpoint of the article.

Response 1: Thanks to the expert teachers for your comments! In the Introduction, a description of other methods for extracting ecological sources is added, and the usefulness of enumerating them is introduced. Its role is to highlight the superiority of the comprehensive identification method. It supplements the relatively new literature in recent years, and initially establishes the knowledge system of the research status in this field.

Point 2: Description of the overview of the research area. It is recommended that the author appropriately add descriptions related to the research content, such as the situation of the forest and grassland in the research area, as well as the current ecological status, in order to better reflect the uniqueness of the research area.

Response 2: Thanks to the expert teachers for your comments! Based on the land use data, the distribution of forest and grassland in the study area is clarified in more detail.  

Point 3: The author is requested to standardize the writing of the paper, strictly modify the format according to the journal's requirements, maintain the alignment of both ends of the entire text, wrap the titles, such as the spacing between title 4.3 and the main text, as well as the clarity of latitude and longitude in the images and annotations.

Response 3: Thanks to the expert teachers for your comments! We have modified the formatting of the article so that it conforms to the requirements. It is reflected in maintaining the alignment of both ends of the full text, and modifying the spacing between the title and the text of 4.3 and its font. The picture has been modified to make the display of latitude and longitude and the picture clearer. Since both the webpage and word itself compress the pictures to a certain extent when uploading the manuscript, we put all the pictures in the attachment and upload them together, hoping to provide clearer pictures for your review. The font size of the latitude and longitude labels has been increased.

Point 4: What are the classification and values of ecological resistance factors in Table 1, and the criteria used by the author for assigning values based on this classification? If it comes from literature review, it is recommended that the author make a supplement.

Response 4: Thanks to the expert teachers for your comments! The criteria for the assignment of ecological factors come from the literature review, that is, the previously cited several papers now numbered 4, 36-38, and added explanations and newer references (now numbered 39-40).

Point 5: In the analysis of the results of ecological service function indicators, the author analyzes them from a time scale. Can we supplement the charts to present the changing trend more clearly; The author listed the spatial distribution maps for the years 2000, 2010, 2015, and 2020. However, the text did not describe each map, and the significance of listing them all is not significant. It is recommended that the author place them in supplementary materials.

Response 5: Thanks to the expert teachers for your comments! Added pictures to more clearly present the changing trend; put the spatial distribution maps for 2000, 2010, 2015, and 2020 into the supplementary material.

Point 6: 4.4 In the results section, it is recommended that the author add the latest references to support the content related to carbon water utilization efficiency and climate regulation, such as Goldenberg R, et al. 2021, Scientific Reports.

Response 6: Thanks to the expert teachers for your comments! Updated references were added in 4.4 to support the content related to carbon water use efficiency and climate regulation.

Point 7: Result analysis should revolve around scientific issues and avoid stacking results. As listed in section 3.2, corresponding conclusions around scientific issues can be added.

Response 7: Thanks to the expert teachers for your comments! The content of 3.2 has been streamlined to make the corresponding conclusions around scientific issues in the last paragraph of each part more prominent.

Point 8: The conclusion is a simple summary from three aspects. Referring to the complex network theory, topological indicators such as Degree distribution and clustering coefficient are calculated, and their correlation with ecological service capability is discussed. The research on the relationship between the two is the focus of this paper, but the conclusion is not accurate enough, and the author is suggested to supplement.

Response 8: Thanks to the expert teachers for your comments! The description of the correlation between each topological index and the ecological service capacity is supplemented and modified to make the conclusion more accurate.

Finally, once again, we would like to express our sincere thanks to you for your valuable comments!

Reviewer 2 Report

The author has carefully revised the paper and the quality has been greatly improved and I think it has met the requirements for publication. But before it can be accepted, several minor issues need to be addressed:

1.      Line 166, between 37°39°N and 108°111°E, revised to between 37°-39° and 108°-111°E.

2.      Why is there no shrub type in the land use type in Figure 3, but there exists shrub type in the ecological resistance factors for land use?

3.      Figure 11, remove the 0 from the horizontal coordinate.

Author Response

Response to Reviewer 2 Comments

Thank you for your careful reading of the manuscript and for these very valuable suggestions. We have made detailed changes to the manuscript based on your suggestions. The line numbers refer to the revised manuscript. The red font in the manuscript is the part that has been revised. If you thinks it is too cluttered in revision mode, you can choose to turn off displaying all marks in word. We have responded to each of your suggestions below.

Point 1: Line 166, between 37°39°N and 108°111°E, revised to between 37°-39° and 108°-111°E.

Response 1: Thanks to the expert teachers for your comments! Line 166, between 37°39°N and 108°111°E, has been revised to read between 37°-39° and 108°-111°E.

Point 2: Why is there no shrub type in the land use type in Figure 3, but there exists shrub type in the ecological resistance factors for land use?

Response 2: Thanks to the expert teachers for your comments! Shrub types in the table of ecological resistance factors for land use (Table 1) have been removed.

Point 3: Figure 11, remove the 0 from the horizontal coordinate.

Response 3: Thanks to the expert teachers for your comments! The "0" on the abscissa in the original Figure 11 (now Figure 10) has been removed and changed to start with "1".

Finally, once again, we would like to express our sincere thanks to you for your valuable comments!

Reviewer 3 Report

We do not have any suggestions.

Author Response

Thank you for your careful reading of the manuscript and for these very valuable suggestions.